# Elucidating the selection mechanisms in context-dependent computation through low-rank neural network modeling

**Yiteng Zhang[1,2], Jianfeng Feng[1,3,4]\*, Bin Min[2]\***

[1]School of Data Science, Fudan University, Shanghai, China; [2]Lingang Laboratory, Shanghai, China; [3]Institute of Science and Technology for Brain-Inspired Intelligence, Fudan University, Shanghai, China; [4]Key Laboratory of Computational Neuroscience and Brain-Inspired Intelligence, Fudan University, Shanghai, China

## eLife Assessment

This study provides an **important** set of analyses and theoretical derivations to understand the mechanisms used by recurrent neural networks (RNNs) to perform context-dependent accumulation of evidence. The results regarding the dimensionality and neural dynamical signatures of RNNs are **convincing** and provide new avenues to study the mechanisms underlying context-dependent computations. This manuscript will be of interest to a broad audience in systems and computational neuroscience.

**\*For correspondence:**
jianfeng64@gmail.com (JF);
minbmath@gmail.com (BM)

## Abstract

Humans and animals exhibit a remarkable ability to selectively filter out irrelevant information based on context. However, the neural mechanisms underlying this context-dependent selection process remain elusive. Recently, the issue of discriminating between two prevalent selection mechanisms—input modulation versus selection vector modulation—with neural activity data has been highlighted as one of the major challenges in the study of individual variability underlying context-dependent decision-making (CDM). Here, we investigated these selection mechanisms through low-rank neural network modeling of the CDM task. We first showed that only input modulation was allowed in rank-one neural networks and additional dimensions of network connectivity were required to endow neural networks with selection vector modulation. Through rigorous information flow analysis, we gained a mechanistic understanding of why additional dimensions are required for selection vector modulation and how additional dimensions specifically contribute to selection vector modulation. This new understanding then led to the identification of novel neural dynamical signatures for selection vector modulation at both single neuron and population levels. Together, our results provide a rigorous theoretical framework linking network connectivity, neural dynamics, and selection mechanisms, paving the way towards elucidating the circuit mechanisms when studying individual variability in context-dependent computation.

## Introduction

Imagine you are playing a card game. Your strategy depends not only on the cards you have but also on what your opponents are doing. As the game progresses, you adjust your moves based on their actions to increase your chances of winning. This example illustrates how much of our decision-making, both in everyday life and in more complex tasks, is influenced by the context in which we are acting (*Miller and Cohen, 2001*; *Roy et al., 2010*; *Mante et al., 2013*; *Saez et al., 2015*; *Siegel et al., 2015*; *Bernardi et al., 2020*; *Takagi et al., 2021*; *Flesch et al., 2022*; *Barbosa et al., 2023*).

However, how the brain performs such context-dependent computation remains elusive (*Fusi et al., 2016*; *Cohen, 2017*; *Badre et al., 2021*; *Okazawa and Kiani, 2023*).

Using a monkey CDM behavioral paradigm, a recent work uncovered a novel mechanism in the brain that helps adjust decisions based on context (*Mante et al., 2013*). This mechanism, called 'selection vector modulation,' is distinct from the early sensory input modulation counterpart (*Desimone and Duncan, 1995*; *Noudoost et al., 2010*). More recently, building on this work, new research with rats supported a novel theoretical framework regarding how the brain makes context-dependent decisions and revealed how this process may vary between individuals (*Pagan et al., 2025*). Critically, this theoretical framework pointed out that current neurophysiological data fell short of distinguishing between selection vector modulation and sensory input modulation, calling for rethinking what kind of evidence is required for differentiating different selection mechanisms.

Here, we aim to address this challenge by using the low-rank recurrent neural network (RNN) modeling approach (*Landau and Sompolinsky, 2018*; *Mastrogiuseppe and Ostojic, 2018*; *Kadmon et al., 2020*; *Schuessler et al., 2020*; *Beiran et al., 2021*; *Beiran et al., 2023*; *Dubreuil et al., 2022*; *Valente et al., 2022*; *Ostojic and Fusi, 2024*). This approach allowed us to simulate and better understand the neural processes behind context-dependent computation. More importantly, endowed by the low-rank RNN theory (*Beiran et al., 2021*; *Dubreuil et al., 2022*), this approach allowed us to develop a set of analyses and derivations uncovering a previously unknown link between connectivity dimensionality, neural dynamics, and selection mechanisms. This link then led to the identification of novel neural dynamical signatures for selection vector modulation at both the single neuron and population levels. Together, our work provides a neural circuit basis for different selection mechanisms, shedding new light on the study of individual variability in neural computation underlying the ubiquitous context-dependent behaviors.

## Results

### Task paradigm, key concept, and modeling approach

The task paradigm we focused on is the pulse-based CDM task (*Pagan et al., 2022*), a novel rat-version CDM paradigm inspired by the previous monkey CDM work (*Mante et al., 2013*). In this paradigm, rats were presented with sequences of randomly-timed auditory pulses that varied in both location and frequency (*Figure 1A*). In alternating blocks of trials, rats were cued by an external context signal to determine the prevalent location (in the 'LOC' context) or frequency (in the 'FRQ' context). Note that compared to the continuous sensory input setting in previous works (e.g. *Mante et al., 2013*), this pulse-based sensory input setting allowed the experimenters to better characterize both behavioral and neural responses (*Pagan et al., 2022*). We will also demonstrate the unique advantage of this pulse-based input setting later in the present study (e.g. Figure 7).

To solve this task, rats had to select the relevant information for the downstream evidence accumulation process based upon the context. There were at least two different mechanisms capable of performing this selection operation, i.e., selection vector modulation and input modulation (*Mante et al., 2013*; *Pagan et al., 2022*). To better introduce these mechanisms, we first reviewed the classical linearized dynamical systems analysis and the concept of selection vector (*Figure 1B*; *Mante et al., 2013*; *Sussillo and Barak, 2013*; *Maheswaranathan et al., 2019*; *Maheswaranathan and Sussillo, 2020*; *Nair et al., 2023*). In the linearized dynamical systems analysis, the neural dynamics around the choice axis (*Figure 1B*, red line) is approximated by a line attractor model. Specifically, the dynamics in the absence of external input can be approximated by the following linear equation $\frac{dr}{dt} = Mr$, where $M$ is a matrix with one eigenvalue being equal to 0 and all other eigenvalues having a negative real part. For brevity, let us denote the left eigenvector of the 0 eigenvalue as $s$. In this linear dynamical system, the effect of any given perturbation can be decomposed along the directions of different eigenvectors: the projection onto the $s$ direction will remain constant while the projections onto all other eigenvectors will exponentially decay to zero. Thus, for any given input $I$, only the component projecting onto the $s$ direction (i.e. $I \cdot s$) can be integrated along the line attractor (see Methods for more details). In other words, $s$ serves as a vector selecting the input information, which is known as the 'selection vector' in the literature (*Mante et al., 2013*).

Two distinct selection mechanisms can then be introduced based upon the concept of selection vector (*Pagan et al., 2022*). Specifically, to perform the CDM task, the stimulus input (LOC input, for

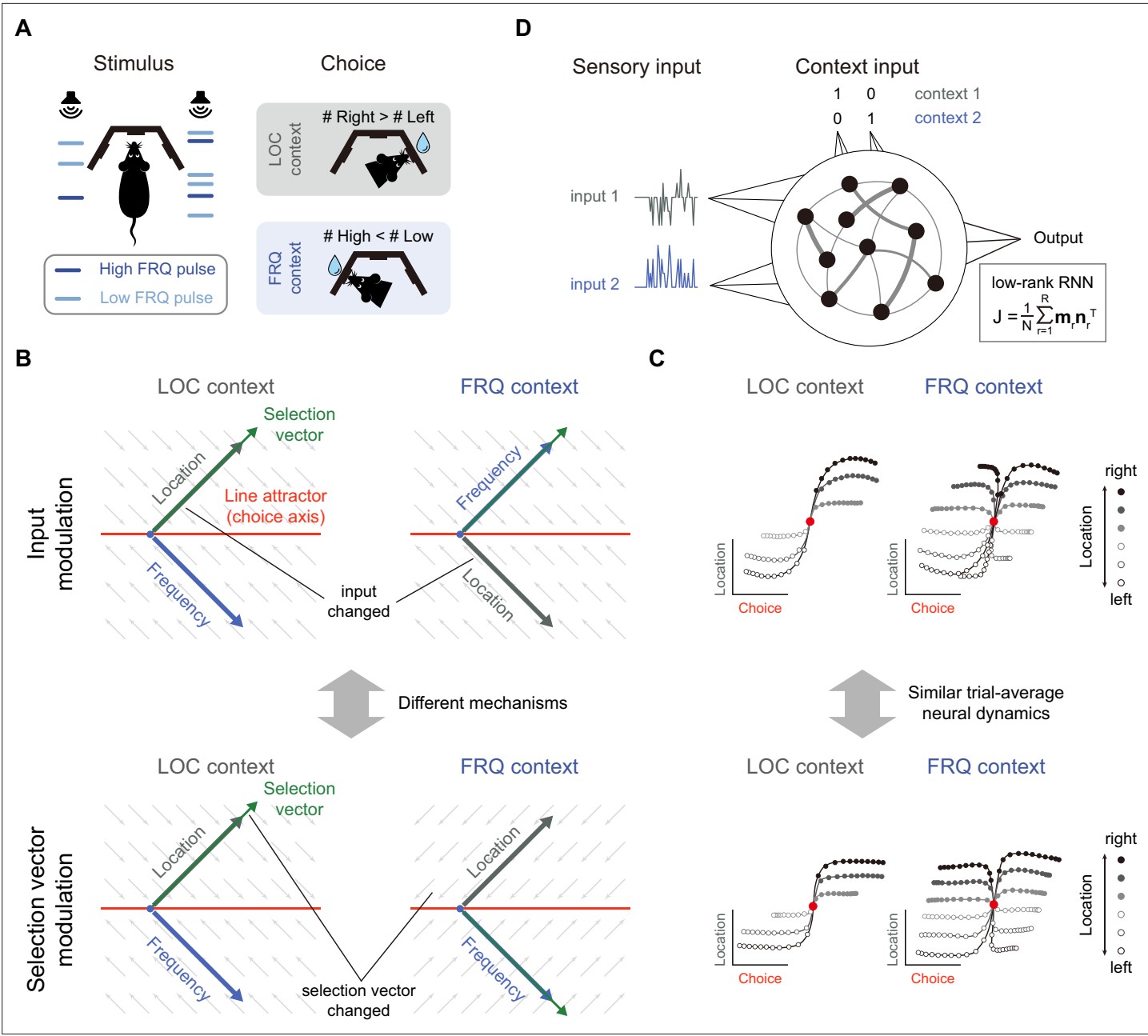

**Figure 1.** Prevalent candidate selection mechanisms in context-dependent decision-making (CDM) cannot be dissociated by classical neural dynamics analysis. (**A**) A pulse-based context-dependent decision-making task (adapted from *Pagan et al., 2022*). In each trial, rats were first cued by sound to indicate whether the current context was the location (LOC) context or the frequency (FRQ) context. Subsequently, rats were presented with a sequence of randomly timed auditory pulses. Each pulse could come from either the left speaker or right speaker and could be of low frequency (6.5 kHz, light blue) or high frequency (14 kHz, dark blue). In the LOC context, rats were trained to turn right (left) if more pulses are emitted from the right (left) speaker. In the FRQ context, rats were trained to turn right (left) if there are more (fewer) high-frequency pulses compared to low-frequency pulses. (**B**) Two prevalent candidate mechanisms for context-dependent decision-making. *Top*: The input modulation mechanism. In this scenario, while the selection vector remains invariant across contexts, the stimulus input representation is altered in a way such that only the relevant stimulus input representation (i.e. the location input in the LOC context and the frequency input in the FRQ context) is well aligned with the selection vector, thereby fulfilling the requirement of context-dependent computation. *Bottom*: The selection vector modulation mechanism. In this scenario, although the stimulus input representation remains constant across different contexts, the selection vector itself is altered by the context input to align with the relevant sensory input. Red line: line attractor (choice axis). Green arrow: selection vector. Thick gray and blue arrows stand for the projections of the location and frequency input representation directions on the space spanned by the line attractor and selection vector, respectively. The small gray arrows stand for direction of relaxing dynamics. (**C**) Networks with distinct selection mechanisms may lead to similar trial-averaged neural dynamics (adapted from *Pagan et al., 2022*). In a model with pure input modulation, the irrelevant sensory input can still be represented by the network in a

*Figure 1 continued on next page*

*Figure 1 continued*

direction orthogonal to the selection vector. Therefore, using the classical targeted dimensionality reduction method (**Mante et al., 2013**), both the input modulation model (top) and the selection vector modulation model (bottom) would exhibit similar trial-averaged neural dynamics as shown in **Pagan et al., 2022**. (**D**) The setting of low-rank RNN modeling for the CDM task. The network has four input channels. Input 1 and input 2 represent two sensory inputs, while the other two channels indicate the context. The connectivity matrix $J$ is constrained to be low-rank, expressed as $\frac{1}{N}\sum_{r=1}^{R} m_r n_r^T$, where $N$ is the number of neurons, $R$ is the matrix's rank, and $m_r n_r^T$ is a rank-1 matrix formed by the outer product of two $N$-dimensional connectivity vectors $m_r$ and $n_r$.

example) must have a larger impact on evidence accumulation in the relevant context (LOC context) than in the irrelevant context (FRQ context). That is, $I \cdot s$ must be larger in the relevant context than in the irrelevant context. The difference between these two can be decomposed into two components:

$$\Delta(I \cdot s) = \Delta I \cdot \bar{s} + \bar{I} \cdot \Delta s, \tag{1}$$

where the $\Delta$ symbol denotes difference across two contexts (relevant – irrelevant) and the bar symbol denotes average across two contexts (see Methods for more details). The first component $\Delta I \cdot \bar{s}$ is called input modulation in which the change of input information across different contexts is emphasized (**Figure 1B**, top). In contrast, the second component $\bar{I} \cdot \Delta s$ is called selection vector modulation in which the change of selection vector across contexts is instead highlighted (**Figure 1B**, bottom).

While these two selection mechanisms were clearly defined, recent work showed that it is actually challenging to differentiate them through neural dynamics (**Pagan et al., 2022**). For example, both input modulation and selection vector modulation can lead to similar trial-averaged neural dynamics through targeted dimensionality reduction (**Figure 1C**; **Pagan et al., 2022**). Take the input modulation as an example (**Figure 1C**, top). One noticeable aspect we can observe is that the input information (e.g. location information) is preserved in both relevant (LOC context) and irrelevant contexts (FRQ context), which seems contradictory to the definition of input modulation. What is the mechanism underlying this counterintuitive result? As Pagan et al. pointed out earlier, input modulation is not the input change ($\Delta I$) per se. Rather, it means the change of input multiplied by selection vector (i.e. $\Delta I \cdot \bar{s}$). Therefore, for the input modulation, while input information indeed is modulated by context along the selection vector direction, input information can still be preserved across contexts along other directions orthogonal to the selection vector, which explains the counterintuitive result and highlights the challenge of distinguishing input modulation from selection vector modulation in experiments (**Pagan et al., 2022**).

In this study, we sought to address this challenge using the low-rank RNN modeling approach. In contrast to the 'black-box' vanilla RNN approach (e.g. **Mante et al., 2013**), the low-rank RNN approach features both well-controlled model complexity and mechanistic transparency, potentially providing a fresh view into the mechanisms underlying the intriguing selection process. Specifically, the low-rank RNNs we studied here implemented an input-output task structure similar to the classical RNN modeling work of CDM (**Figure 1D**; **Mante et al., 2013**). More concretely, the hidden state $x$ of a low-rank RNN with $N$ neurons evolves over time according to

$$\tau \frac{dx}{dt} = -x + J\phi(x) + \sum_{s=1}^{2} I_s u_s(t) + \sum_{s=1}^{2} I_s^{ctx} u_s^{ctx}(t) \tag{2}$$

where $J = \sum_{r=1}^{R} m_r n_r^T / N$ is a low-rank matrix with $R$ output vectors $m_r, r = 1, \cdots, R$ and R input-selection vectors $n_r, r = 1, \cdots, R$, $\tau$ is the time constant of single neurons, $\phi$ is the nonlinear activation function, $u_s(t), s = 1, 2$ embedded into the network through $I_s$ mimic the location and frequency click inputs, and $u_s^{ctx}(t), s = 1, 2$ embedded through $I_s^{ctx}$ indicate whether the current context is location or frequency. The output of the network is a linear projection of neural activity (see Methods for more model details). Under this general architectural setting, on one hand, through controlling the rank $R$ of the matrix $J$ during backpropagation training, we can determine the minimal rank required for performing the CDM task and reverse-engineer the underlying mechanism, which will be demonstrated in **Figure 2**. On the other hand, recent theoretical progress of low-rank RNNs (**Beiran et al., 2021**; **Beiran et al., 2021**; **Dubreuil et al., 2022**) enabled us to explicitly construct neural network models with mechanistic transparency, complementing the reverse-engineering analysis (**Mante et al., 2013**; **Sussillo and Barak, 2013**), which will be shown in **Figure 3**.

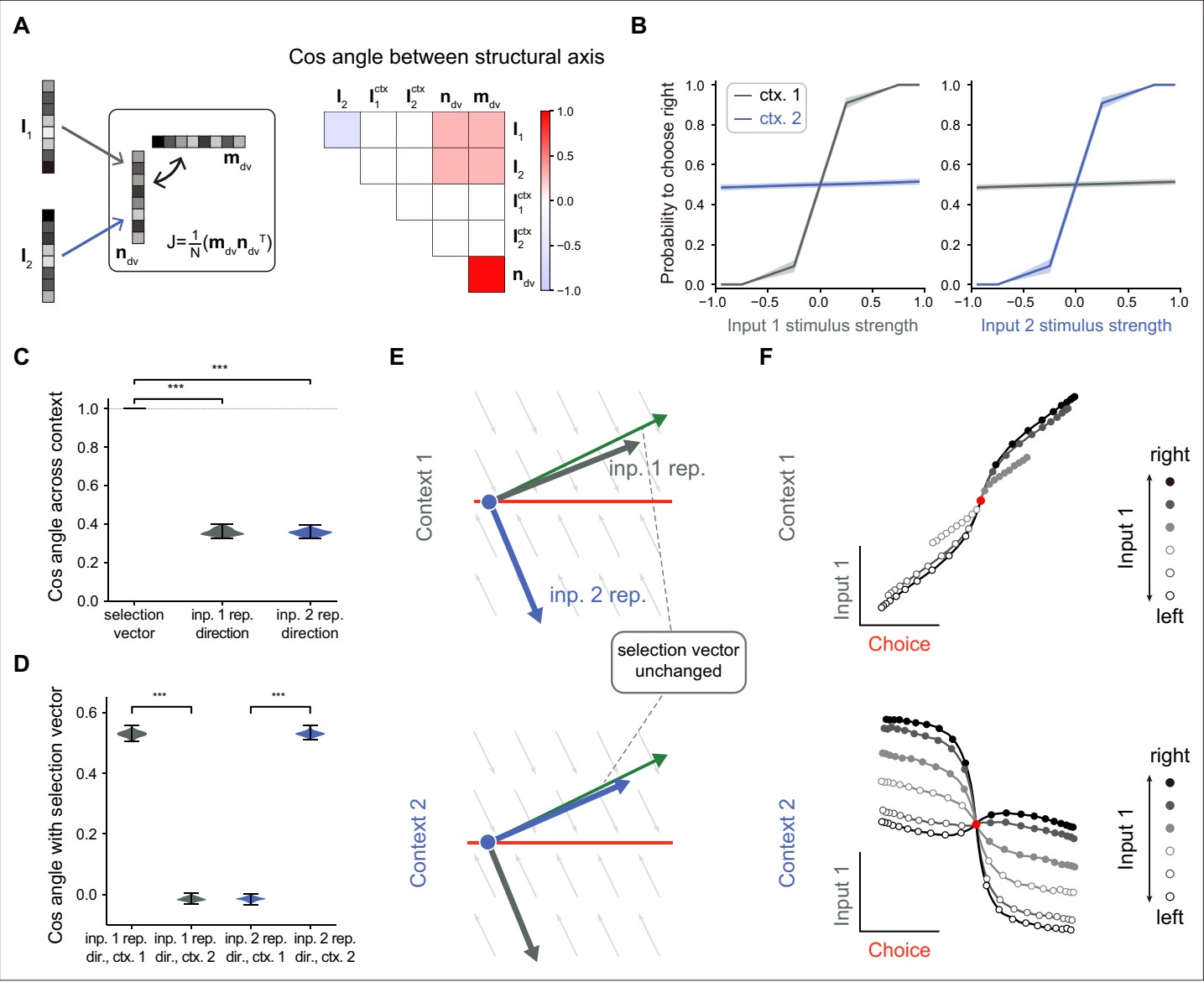

**Figure 2.** No selection vector modulation in rank-1 neural network models. (**A**) Illustration of rank-1 connectivity matrix structure. *Left*: a rank-1 matrix can be represented as the outer product of an output vector $\boldsymbol{m}_{dv}$ and an input-selection vector $\boldsymbol{n}_{dv}$, of which the input-selection vector $\boldsymbol{n}_{dv}$ played the role of selecting the input information through its overlap with the input embedding vectors $\boldsymbol{I}_1$ and $\boldsymbol{I}_2$. The context signals are fed forward to the network with embedding vectors $\boldsymbol{I}_1^{ctx}$ and $\boldsymbol{I}_2^{ctx}$. Since the overlap between the context embedding vectors and input-selection vector $\boldsymbol{n}_{dv}$ are close to 0, for simplicity, we omitted the context embedding vectors here. *Right*: an example of the trained rank-1 connectivity structure characterized by the cosine angle between every pair of connectivity vectors (see *Figure 2—figure supplement 1* and Methods for details). (**B**) The psychometric curve of the trained rank-1 recurrent neural networks (RNNs). In context 1, input 1 strongly affects the choice, while input 2 has little impact on the choice. In context 2, the effect of input 1 and input 2 on the choice is exchanged. The shaded area indicates the standard deviation. Ctx. 1, context 1. Ctx. 2, context 2. (**C**) Characterizing the change of selection vector as well as input representation direction across contexts using cosine angle. The selection vector in each context is computed using linearized dynamical system analysis. The input representation direction is defined as the elementwise multiplication between the single neuron gain vector and the input embedding vector (see Methods for details). \*\*\*p<0.001, one-way ANOVA test, n=100. Inp., input. Rep., representation. (**D**) Characterizing the overlap between the input representation direction and the selection vector. \*\*\*p<0.001, one-way ANOVA test, n=100. Dir., direction. (**E**) The state space analysis, for example, trained rank-1 RNN. The space is spanned by the line attractor axis (red line) and the selection vector (green arrow). (**F**) Trial-averaged dynamics for example rank-1 RNN. We applied targeted dimensionality reduction (TDR) to identify the choice, input 1, and input 2 axes. The neuron activities were averaged according to input 1 strength, choice and context, and then projected onto the choice and input 1 axes to obtain the trial-averaged population dynamics (see Methods for details).

The online version of this article includes the following figure supplement(s) for figure 2:

**Figure supplement 1.** Connectivity structure for the example rank-1 recurrent neural network (RNN).

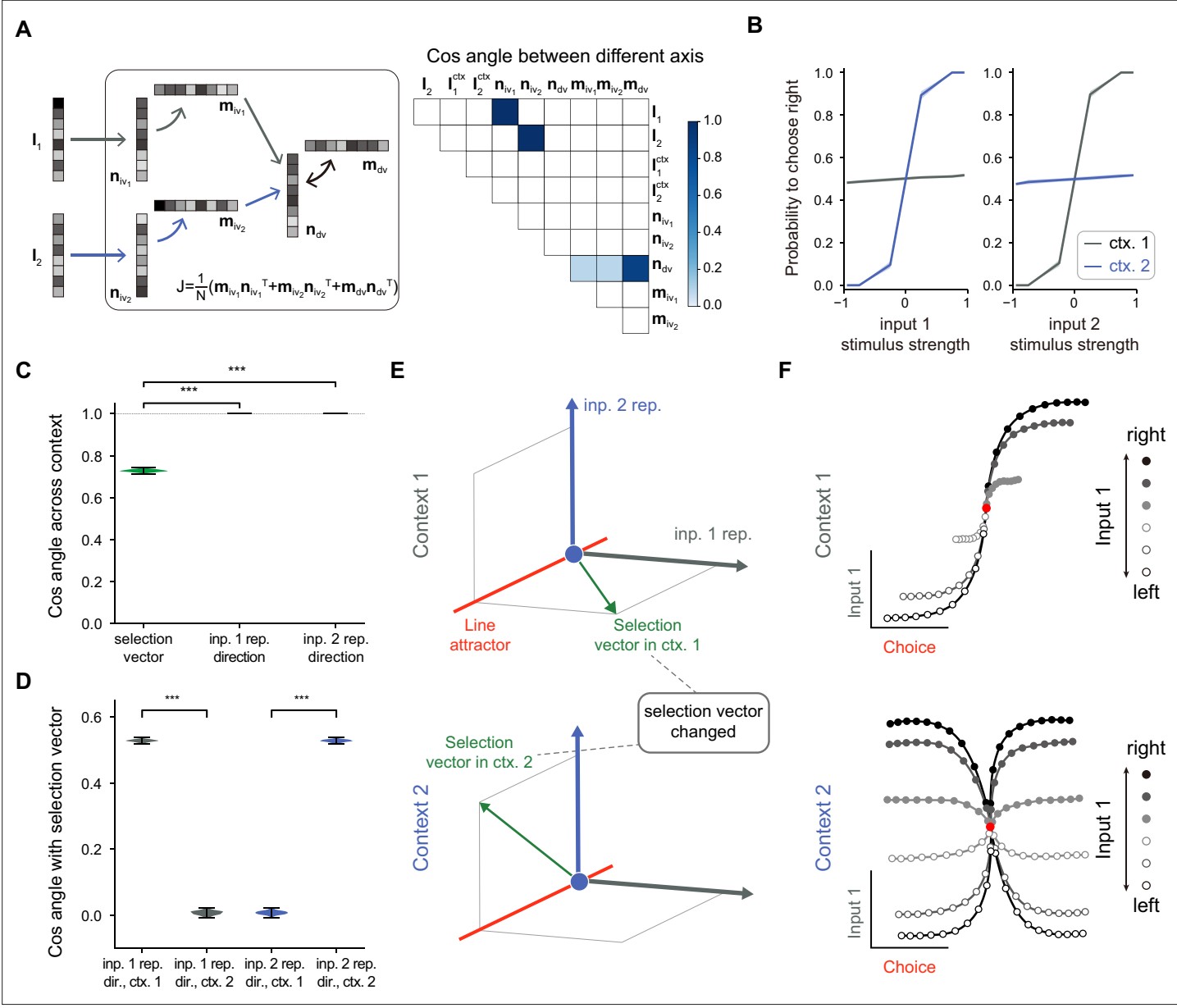

**Figure 3.** A rank-3 neural network model with pure selection vector modulation. (**A**) Illustration of the utilized rank-3 connectivity matrix structure. *Left*: the rank-3 matrix can be represented as the summation of three outer products, including the one with the output vector $m_{dv}$ and the input-selection vector $n_{dv}$, the one with the output vector $m_{iv_1}$ and the input-selection vector $n_{iv_1}$, and the one with the output vector $m_{iv_2}$ and the input-selection vector $n_{iv_2}$, of which the input-selection vectors $n_{iv_1}$ and $n_{iv_2}$ played the role of selecting the input information from $I_1$ and $I_2$, respectively. *Right*: the connectivity structure of the handcrafted RNN model characterized by the cosine angle between every pair of connectivity vectors (see ***Figure 3—figure supplement 1*** and Methods for more details). (**B**) The psychometric curve of the handcrafted rank-3 recurrent neural network (RNN) model. (**C**) Characterizing the change of selection vector as well as input representation direction across contexts using cosine angle. The selection vector in each context is computed using linearized dynamical system analysis. The input representation direction is defined as the elementwise multiplication between the single neuron gain vector and the input embedding vector (see Methods for details). ***$p<0.001$, one-way ANOVA test, $n=100$. (**D**) Characterizing the overlap between the input representation direction and the selection vector. ***$p<0.001$, one-way ANOVA test, $n=100$. (**E**) The state space analysis for example rank-3 RNN. The space is spanned by the line attractor axis (red line, invariant across contexts), selection vector in context 1 (green arrow, top panel), and selection vector in context 2 (green arrow, bottom panel). (**F**) Trial-averaged dynamics for example rank-3 RNN.

The online version of this article includes the following figure supplement(s) for figure 3:

**Figure supplement 1.** Connectivity structure for the example rank-3 recurrent neural network (RNN).

## No selection vector modulation in rank-one models

In the literature, it was found that rank-one RNN models suffice to solve the CDM task (*Dubreuil et al., 2022*). Here, we further asked whether selection vector modulation can occur in rank-one RNN models. To this end, we trained many rank-1 models (see Methods for details) and found that indeed having rank-1 connectivity (e.g. with the overlap structure listed in *Figure 2A*; for the detailed connectivity structure, see *Figure 2—figure supplement 1*) is sufficient to perform the CDM task, consistent with the earlier work. As shown in *Figure 2B*, in context 1, the decision was made based on input-1 evidence, ignoring input-2 evidence, indicating that the network can effectively filter out irrelevant information. To answer what kind of selection mechanisms underlie this context-dependent computation, we computed the selection vector in two contexts through linearized dynamical systems analysis (*Sussillo and Barak, 2013*). Cosine angle analysis revealed that selection vectors kept invariant across different contexts (*Figure 2C*, left), indicating no selection vector modulation. This result was preserved across different hyperparameter settings (such as different regularization coefficients or activation functions). Note that this result actually can be mathematically proved (*Pagan et al., 2023*; *Pagan et al., 2025*). Therefore, our modeling result reconfirmed the limitation of rank-one models on selection vector modulation.

While the selection vector was not altered by contexts, the direction of input representations changed significantly across different contexts (*Figure 2C*, left; see Methods for the definition of input representation direction). Further analysis revealed that the overlap between the input representation direction and the unchanged selection vector is large in the relevant context and small in the irrelevant context, supporting the input modulation mechanism (*Figure 2D*). These results indicate that while a rank-1 network can perform the task, it can only achieve flexible computation through input modulation (*Figure 2E*). Importantly, when applying a similar targeted dimensionality reduction method to this rank-1 model, we found that the irrelevant sensory input information was indeed well-represented in neural activity state space (*Figure 2F*), supporting the conclusion made in the Pagan et al. paper that the presence of irrelevant sensory input in neural state space cannot be used as a reliable indicator for the absence of input modulation (*Figure 1C*).

In summary, we conclude that to study the mechanism of selection vector modulation, instead of limiting to the simplest model of CDM task, it is necessary to explore network models with higher ranks.

## A low-rank model with pure selection vector modulation

To study the mechanism of selection vector modulation, we designed a rank-3 neural network model, with one additional rank for each sensory input feature (i.e. $\boldsymbol{m}_{iv_1}\boldsymbol{n}_{iv_1}^T$ for input 1 and $\boldsymbol{m}_{iv_2}\boldsymbol{n}_{iv_2}^T$ for input 2; *Figure 3A*, left). Specifically, we ensured that $\boldsymbol{I}_1(\boldsymbol{I}_2)$ has a positive overlap with $\boldsymbol{n}_{iv_1}(\boldsymbol{n}_{iv_2})$ and zero overlap with $\boldsymbol{n}_{dv}$, while $\boldsymbol{m}_{iv1}(\boldsymbol{m}_{iv_2})$ has a positive overlap with $\boldsymbol{n}_{dv}$ (*Figure 3A*, right; see *Figure 3—figure supplement 1* and Methods for more details). Moreover, our rank-3 network relies on a multi-population structure, consistent with the notion that higher-rank networks still require a multi-population structure to perform flexible computations (*Dubreuil et al., 2022*). This configuration implies that the stimulus input 1 (2) is first selected by $\boldsymbol{n}_{iv_1}(\boldsymbol{n}_{iv_2})$, represented by $\boldsymbol{m}_{iv_1}(\boldsymbol{m}_{iv_2})$, and subsequently selected by $\boldsymbol{n}_{dv}$ before being integrated by the accumulator. In principle, this sequential selection process enables more sophisticated contextual modulations.

We confirmed that a model with such a connectivity structure can perform the task (*Figure 3B*) and then conducted an analysis similar to that performed for rank-1 models. Unlike the rank-1 model, the selection vector for this rank-3 model changes across contexts while the input representation direction remains invariant (*Figure 3C*). Further analysis revealed that the overlap between the selection vector and the unchanged input representation direction is large in the relevant context and small in the irrelevant context (*Figure 3D*), supporting a pure selection vector modulation mechanism (*Figure 3E*) distinct from the input modulation counterpart shown in *Figure 2D*. When applying a similar targeted dimensionality reduction method to this rank-3 model, as what we expected, we found that both relevant and irrelevant sensory input information was indeed well-represented in neural activity state space (*Figure 3F*), which was indistinguishable from the input modulation counterpart (*Figure 2F*).

Together, through investigating these two extreme cases—one with pure input modulation and the other with pure selection vector modulation, we not only reconfirm the challenge of distinguishing input modulation from selection modulation based on neural activity data (*Pagan et al., 2022*) but

also point out the previously unknown link between selection vector modulation and network connectivity dimensionality.

## Understanding context-dependent modulation in Figs. 2 and 3 through pathway-based information flow analysis

What is the machinery underlying this link between selection vector modulation and network connectivity dimensionality? One possible way to address this issue is through linearized dynamical systems analysis: first computing the selection vector and the sensory input representation direction through reverse-engineering (*Sussillo and Barak, 2013*) and then calculating both selection vector modulation and input modulation according to *Equation 1*. However, the connection between the network connectivity dimensionality and the selection vector obtained through reverse-engineering is implicit and in general non-trivial (*Pagan et al., 2023*), hindering further investigation of the underlying machinery. Here, by combining recent theoretical progress in low-rank RNNs (*Mastrogiuseppe and Ostojic, 2018*; *Dubreuil et al., 2022*) and linearized dynamical systems analysis (*Sussillo and Barak, 2013*), we introduced a novel pathway-based information flow analysis approach, providing an explicit link between network connectivity, neural dynamics, and selection mechanisms.

To start with, the low-rank RNN dynamics (i.e. *Equation 2*) can be described by an information flow graph, with each task variable as a node and each effective coupling between task variables as an edge (*Mastrogiuseppe and Ostojic, 2018*; *Dubreuil et al., 2022*). Take the rank-1 RNN in *Figure 2* as an example. A graph with three nodes, including two input variables $\kappa_{p_s}(t)$, $s = 1, 2$ and one decision variable $\kappa_{dv}(t)$, suffices (*Figure 4A*; see Methods for more details). In this graph, the dynamical evolution of the task variable $\kappa_{dv}$ can be expressed as:

$$\tau \frac{d\kappa_{dv}}{dt} = -\kappa_{dv} + E_{inp_1 \to dv}\kappa_{inp_1} + E_{inp_2 \to dv}\kappa_{inp_2} + E_{dv \to dv}\kappa_{dv} \tag{3}$$

where the effective coupling $E_{inp_s \to dv}$ from the input variable $\kappa_{inp_s}$ to the decision variable $\kappa_{dv}$ is equal to the overlap between the input representation direction $\widetilde{I}_s$ (each element is defined by $\widetilde{I}_{s,i} = \phi'(x_i) I_{s,i}$) and the input-selection vector $n_{dv}$. More precisely, $E_{inp_s \to dv} = \left\langle \widetilde{I}_s, n_{dv} \right\rangle$, where $\langle a, b \rangle$ is defined as $\frac{1}{N}\sum_1^N a_i b_i$ for two length-$N$ vectors. Since the input representation direction $\widetilde{I}_s$ depends on the single neuron gain $\phi'$ and the context input can modulate this gain, the effective coupling $E_{inp_s \to dv}$ is context-dependent. Indeed, as shown in *Figure 4A*, *Figure 4—figure supplement 1A*, $E_{inp_1 \to dv}$ exhibited a large value in context 1 but was negligible in context 2, while $E_{inp_2 \to dv}$ exhibited a large value in context 2 but was negligible in context 1. In other words, information from the input variable can arrive at the decision variable only in the relevant context, which exactly is the computation required by the CDM task.

To get a more intuitive understanding of the underlying information flow process, we discretized the equation and followed the information flow step by step. Specifically, by discretizing *Equation 3* using Euler's method with a time step equal to the time constant $\tau$ of the system, we get

$$\kappa_{dv}(t + \tau) = E_{inp_1 \to dv}\kappa_{inp_1}(t) + E_{inp_2 \to dv}\kappa_{inp_2}(t) + E_{dv \to dv}\kappa_{dv}(t) \tag{4}$$

Take context 1 as an example (*Figure 4B*, top panel). Initially, there is no information in the system. In step 1, pulse inputs of size $A_1$ and $A_2$ are placed in the $p_1$ and $p_2$ slots, respectively. The information from these slots, after being multiplied by the corresponding effective coupling, then flows to the $dv$ slot. In context 1, $E_{inp_2 \to dv} \approx 0$, meaning only the content from the $p_1$ slot can arrive at the $dv$ slot. Consequently, in step 2, the information content in the $dv$ slot would be $E_{inp_1 \to dv}A_1$. The following steps will replicate step 2 due to the recurrent connectivity of the $dv$ slot. The scenario in context 2 is similar to context 1, except that only the content from the $p_2$ slot arrives at the $dv$ slot (*Figure 4B*, bottom panel). The continuous dynamics for each task variable given pulse input is displayed in *Figure 4—figure supplement 2A–C*.

The same pathway-based information flow analysis can also be applied to the rank-3 model (*Figure 4C and D*). In this model, similar to the rank-1 models, there are input variables ($k_{inp_1}$ and $k_{inp_2}$) and decision variable ($k_{dv}$). Additionally, it includes intermediate variables ($k_{iv_1}$ and $k_{iv_2}$) corresponding to the activity along the $m_{iv_1}$ and $m_{iv_2}$ axes. In this scenario, instead of flowing directly from the input to the decision variable, the information flows first to the intermediate variables and then to the decision

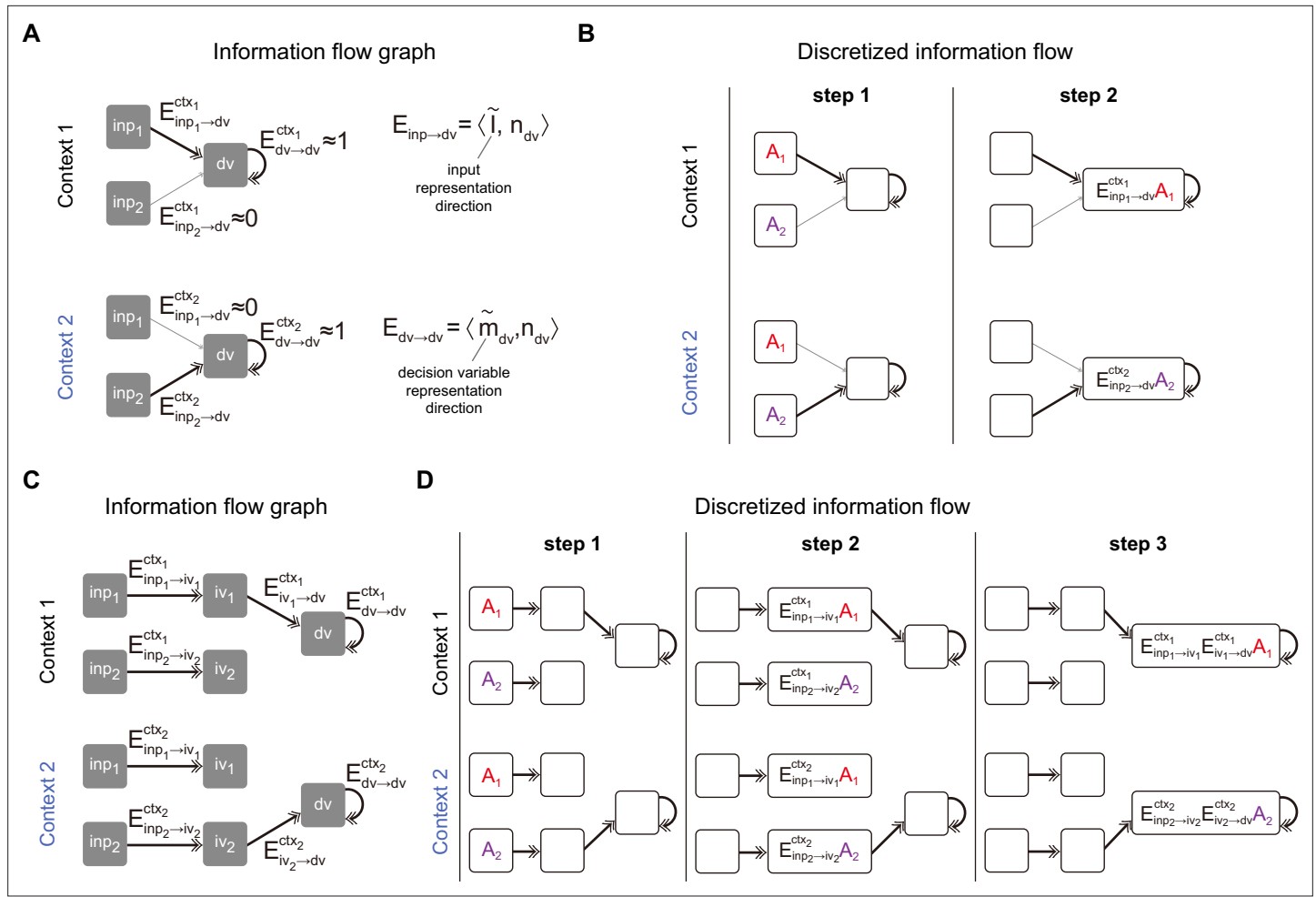

**Figure 4.** Pathway-based information flow analysis. (**A**) The information flow graph of the rank-1 model presented in *Figure 2*. In this graph, nodes represented task variables communicating with each other through directed connections (denoted as $E_{sender \to receiver}$) between them. Note that $E_{sender \to receiver}$ is the overlap between the representation direction of the sender variable (e.g. the representation directions of input variable and decision variable $\tilde{I}_{inp}$ and $\tilde{m}_{dv}$) and the input-selection vector of the receiver variable (e.g. the input-selection vector of decision variable $n_{dv}$). As such, $E_{sender \to receiver}$ naturally inherits the context dependency from the representation direction of task variable: while $E_{inp_1 \to dv}$ exhibited a large value and $E_{inp_2 \to dv}$ was negligible in context 1, the values of these two exchanged in context 2. (**B**) Illustration of information flow dynamics in (**A**) through discretized steps. At step 1, sensory information $A_1$ and $A_2$ were placed in *inp*1 and *inp*2 slots, respectively. Depending on the context, different information contents (i.e. $E_{inp_1 \to dv}A_1$ in context 1 and $E_{inp_2 \to dv}A_2$ in context 2) entered into the $dv$ slot at step 2 and were maintained by recurrent connections in the following steps, which is desirable for the context-dependent decision-making task. (**C**) The information flow graph of the rank-3 model presented in *Figure 3*. Different from (**A**), here to arrive at the $dv$ slot, the input information has to first go through an intermediate slot (e.g. the $p_1 \to iv_1 \to dv$ pathway in context 1 and the $p_2 \to iv_2 \to dv$ pathway in context 2). (**D**) Illustration of information flow dynamics in (**C**) through discretized steps.

The online version of this article includes the following figure supplement(s) for figure 4:

**Figure supplement 1.** Effective coupling between task variables for rank-1 and rank-3 recurrent neural networks (RNNs).

**Figure supplement 2.** Neural activity and task variable dynamics for single pulse input.

variable. These intermediate variables act as intermediate nodes. Introducing these nodes does more than simply increase the steps from the input variable to the decision variable. In the rank-1 case, the context signals can only modulate the pathway from the input to the decision variable. However, in the rank-3 case, context signals can modulate the system in two ways: from the input to the intermediate variables and from the intermediate variables to the decision variable.

Take the rank-3 model introduced in *Figure 3* as an example. Context signals did not alter the representation of input signals, leading to constant effective couplings (i.e. constant $E_{inp_1 \to iv_1}$ and $E_{inp_2 \to iv_2}$) from input to intermediate variables across contexts. Instead, it changed the effective

coupling from the intermediate variables to the decision variable (i.e. large $E_{iv_1 \to dv}$ in context 1 and near zero $E_{iv_1 \to dv}$ in context 2; *Figure 4C*, *Figure 4—figure supplement 1B*). Consider the context 1 scenario in the discrete case. In step 1, pulse inputs of size $A_1$ and $A_2$ are placed in the $p_1$ and $p_2$ slots, respectively. In step 2, information flows to the intermediate slots, with $E_{inp_1 \to iv_1} A_1$ in the $iv_1$ slot and $E_{inp_2 \to iv_2} A_2$ in the $iv_2$ slot. In step 3, only the information in the $iv_1$ slot flows to the $dv$ slot, with the information content being $E_{inp_1 \to iv_1} E_{iv_1 \to dv} A_1$ (*Figure 4D*, top panel). The scenario in context 2 is similar to context 1, except that only the content from the $iv_2$ slot reaches the $dv$ slot in the third step, with the content being $E_{inp_2 \to iv_2} E_{iv_2 \to dv} A_2$ (*Figure 4D*, bottom panel). The continuous dynamics for each task variable given pulse input is displayed in *Figure 4—figure supplement 2D, E*.

Together, this pathway-based information flow analysis provides an in-depth understanding of how the input information can be routed to the accumulator depending on the context, laying the foundation for a novel pathway-based information flow definition of selection vector and input contextual modulations.

## Information flow-based definition of selection vector modulation and selection vector for more general cases

Based on the understanding gained from the pathway-based information flow analysis, we now provide a novel definition of input modulation and selection vector modulation distinct from the one in *Equation 1*. To begin with, we first considered a model with mixed input and selection vector modulation (*Figure 5*, left), instead of studying extreme cases (i.e. one with pure input modulation in *Figure 2* and the other with pure selection vector modulation in *Figure 3*). In this more general model, input information can either go directly to the decision variable (with effective coupling $E_{inp \to dv}$) or first pass through the intermediate variables before reaching the decision variable (with effective coupling $E_{inp \to iv}$ and $E_{iv \to dv}$ respectively). Applying the same information flow analysis, we see that a pulse input of unit size will ultimately reach the $dv$ slot with a magnitude of $E_{inp \to dv} + E_{inp \to iv} E_{iv \to dv}$ (*Figure 5A*, right; see Methods for more details). In other words, the total effective coupling $E_{tol}$ from the input to the decision variable is equal to $E_{inp \to dv} + E_{inp \to iv} E_{iv \to dv}$. Now, it is straightforward to decompose the context-dependent modulation of $E_{tol}$ in terms of input or selection vector change:

$$\Delta E_{tol} = \left( \Delta E_{inp \to dv} + \Delta E_{inp \to iv} \bar{E}_{iv \to dv} \right) + \left( \bar{E}_{inp \to iv} \Delta E_{iv \to dv} \right), \tag{5}$$

in which the first component $\Delta E_{inp \to dv} + \Delta E_{inp \to iv} \bar{E}_{iv \to dv}$ stands for the change of input representation (termed as input modulation) and the second component $\bar{E}_{inp \to iv} \Delta E_{iv \to dv}$ is the one without changing the stimulus input representation (termed as selection vector modulation).

We then asked if this pathway-based definition is equivalent to the one in *Equation 1* based on linearized dynamical system analysis (*Figure 5*; *Pagan et al., 2022*). To answer this question, we numerically compared these two definitions using a family of models with both input and selection vector modulations (*Figure 5C*; see Methods for model details) and found that these two definitions produced the same proportion of selection vector modulation across a wide range of parameter regimes (*Figure 5D*). Together with theoretical derivation of equivalence (see Methods), this consistency confirmed the validity of our pathway-based definition of contextual modulation decomposition.

Having elucidated the pathway-based definitions of input and selection vector modulation, we next provide a novel pathway-based definition of selection vector. For the network depicted in *Figure 5A*, the total effective coupling $E_{inp \to dv} + E_{inp \to iv} E_{iv \to dv}$ can be rewritten as $\left\langle \widetilde{I}, n_{tol} \right\rangle$, where $\widetilde{I}$ is the input representation direction and $n_{tol} = n_{dv} + \left\langle \widetilde{m}_{iv}, n_{dv} \right\rangle n_{iv}$. This reformulation aligns with the insight that the amount of input information that can be integrated by the accumulator is determined by the dot product between the input representation direction $\widetilde{I}$ and the selection vector. Thus, $n_{tol}$ is the selection vector of the circuit in *Figure 5A*.

To better understand why selection vector has such a formula, we visualized the information propagation from input to the choice axis using a low-rank matrix (*Figure 6A*). Specifically, it comprises two components, each corresponding to a distinct pathway. For the first component, input information is directly selected by $n_{dv}$. For the second component, input information is sent first to the intermediate variable and then to the decision variable. This pathway involves two steps: in the first step, the input representation vector is selected by $n_{iv}$, and in the second step, to arrive at the choice axis, the

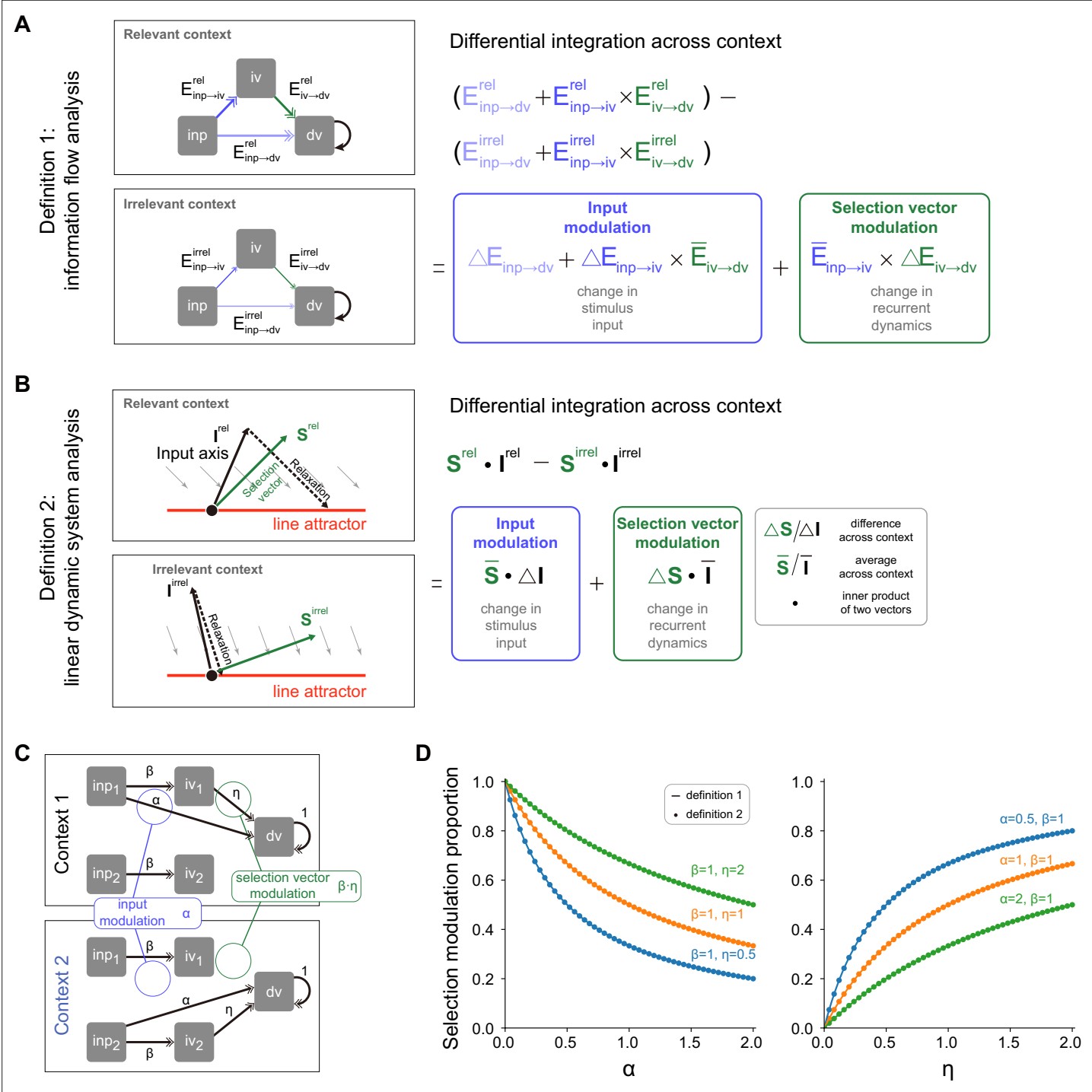

**Figure 5.** A novel pathway-based definition of selection vector modulation. (**A**) A pathway-based decomposition of contextual modulation in a model with both input and selection vector modulations. This definition is based on an explicit formula of the effective connection from the input variable to the decision variable in the model (i.e. $E_{inp \to dv} + E_{inp \to iv}E_{iv \to dv}$; see Method for details). The input modulation component is then defined as the modulation induced by the change of the input representation direction across contexts. The remaining component is then defined as the selection vector modulation one. (**B**) Illustration of contextual modulation decomposition introduced in *Pagan et al., 2022*. In this definition, the selection vector has to be first reverse-engineered through linearized dynamical systems analysis. The input modulation component is then defined as the modulation induced by the change of input representation direction across contexts, while the selection vector modulation component is defined as the one induced by the change of the selection vector across contexts. (**C**) A family of handcrafted recurrent neural networks (RNNs) with both input and selection vector modulations. $\alpha$, $\beta$, and $\eta$ represent the associated effective coupling between task variables. In this model family, the $inp \to dv$ pathway, susceptible to the input modulation, is parameterized by $\alpha$ while the $inp \to iv \to dv$ pathway, susceptible to the selection vector modulation,

*Figure 5 continued on next page*

*Figure 5 continued*

is parameterized by $\beta$ and $\eta$. As such, the ratio of the input modulation to the selection vector modulation can be conveniently controlled by adjusting $\alpha$, $\beta$, and $\eta$. (**D**) Comparison of pathway-based definition in (**A**) with the classical definition in (**B**) using the model family introduced in (**C**).

selected information has to be multiplied by the effective coupling $E_{iv \to dv} = \left\langle \widetilde{m}_{iv}, n_{dv} \right\rangle$. By concatenating these two steps, information propagation from the input to the choice axis in this pathway can be effectively viewed as a selection process mediated by the vector $\left\langle \widetilde{m}_{iv}, n_{dv} \right\rangle n_{iv}$ (termed as the second-order selection vector component). Therefore, $n_{tol}$ provides a novel pathway-based definition of selection vector in the network. We further verified the equivalence between this pathway-based definition and the linearized-dynamical-systems-based classical definition (*Mante et al., 2013*) in our simple circuit through theoretical derivation (see Methods) and numerical comparison (*Figure 6B*).

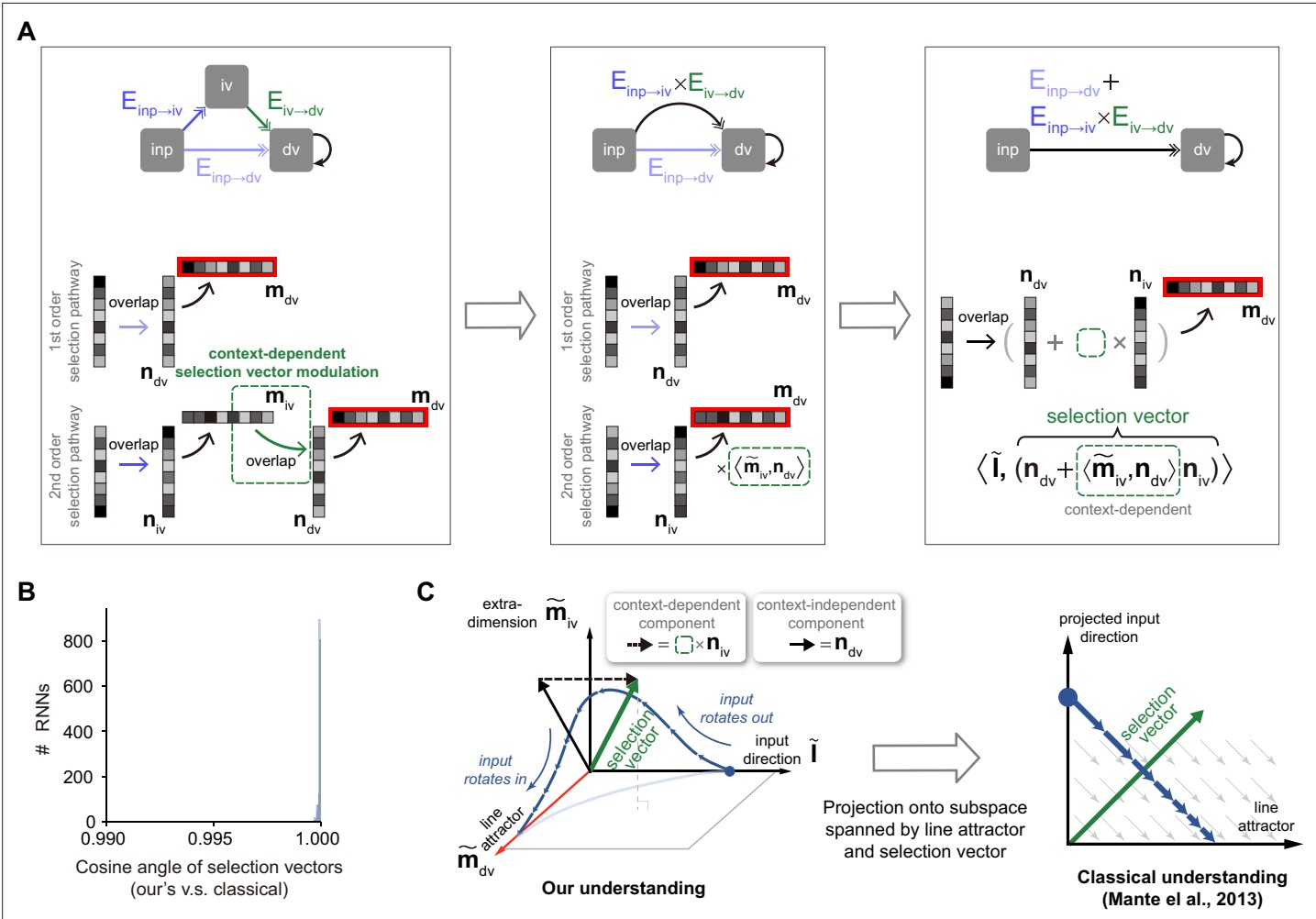

**Figure 6.** An explicit pathway-based formula of selection vector. (**A**) Illustration of how an explicit pathway-based formula of selection vector is derived. In a model with both the first-order selection pathway (i.e. $inp \to dv$) and the second-order selection pathway (i.e. $inp \to iv \to dv$), the second-order pathway can be reduced to a pathway with the effective selection vector $\left\langle \widetilde{m}_{iv}, n_{dv} \right\rangle n_{iv}$ that exhibited the contextual dependency missing in rank-1 models. (**B**) Comparison between this pathway-based selection vector and the classical one (*Mante et al., 2013*) using 1000 recurrent neural networks (RNNs) (see Methods for details). (**C**) The connection between our understanding and the classical understanding in neural state space. Based upon the explicit formula of selection vector in (**A**), the selection vector modulation has to rely on the contextual modulation of additional representation direction (i.e. $\widetilde{m}_{iv}$) orthogonal to both the input representation direction ($\widetilde{I}_{inp}$) and decision variable representation direction ($\widetilde{m}_{dv}$, line attractor). Therefore, it requires at least three dimensions (i.e. $\widetilde{I}_{inp}$, $\widetilde{m}_{dv}$, and $\widetilde{m}_{iv}$) to account for the selection vector modulation in neural state space.

To visualize the pathway-based selection vector in neural activity state space, we found that a minimum of three dimensions is required, including the input representation direction, the decision variable representation direction, and the intermediate variable representation direction (*Figure 6C*, left). This geometric visualization highlighted the role of extra dimensions beyond the classical two-dimensional neural activity space spanned by the line attractor and selection vector (*Figure 6C*, right) in accounting for the selection vector modulation. This is simply because only the second-order selection vector component, which depends on the existence of the intermediate variable, is subject to contextual modulation. In other words, without extra dimensions to support intermediate variable encoding, there will be no selection modulation.

Together, this set of analyses provided a parsimonious pathway-based understanding for both selection vector and its contextual modulation.

## Model prediction verification with vanilla RNN models

The new insights obtained from our new framework enable us to generate testable predictions for better differentiating selection vector modulation from input modulation, a major challenge unresolved in the field (*Pagan et al., 2022*).

First, we predict that it is more likely to have a large proportion of selection vector modulation for a neural network with high-dimensional connectivity. To better explain the underlying rationale, we can simply compare the number of potential connections contributing to the input modulation with those contributing to the selection vector modulation for a given neural network model. For example, in the network presented in *Figure 5*, there are three connections (including light blue, dark blue, and green ones) while only one connection (i.e. the green one) supporting selection vector modulation. For a circuit with many higher-order pathways (e.g. *Figure 7A*), only those connections with the input as the sender is able to support input modulation. In other words, there exist far more connections potentially eligible to support the selection vector modulation (*Figure 7B*), thereby leading to a large selection vector modulation proportion. We then tested this prediction on vanilla RNNs trained through backpropagation (*Figure 7C*; *Mante et al., 2013*; *Song et al., 2016*; *Yang and Wang, 2020*). Using effective dimension (see Methods for a formal definition; *Rudelson and Vershynin, 2007*; *Sanyal et al., 2020*) to quantify the dimensionality of the connectivity matrix, we found a strong positive correlation between the effective dimension of connectivity matrix and the selection vector modulation (*Figure 7D*, left panel and *Figure 7—figure supplement 1A*, see Method for details).

We then asked if we could generate predictions to quantify the proportion of selection vector modulation purely based on neural activities. To this end, as what has been performed in Pagan et al., we took advantage of the pulse-based sensory input setting to calculate the single neuron response kernel (see Methods for details). For a given neuron, the associated response kernel is defined to characterize the influence of a pulse input on its firing rate in later times. For example, for a neuron encoding the decision variable, the response kernel should exhibit a profile accumulating evidence over time (*Figure 7E*, top left). In contrast, for a neuron encoding the sensory input, the response kernel should exhibit an exponential decay profile with time constant $\tau$ (*Figure 7E*, top right). For each RNN trained through backpropagation, we then examined the associated single neuron response kernels. We found that for the model with low effective dimension (denoted by a star marker in *Figure 7D*), there were mainly two types of response kernels, including sensory input profile and decision variable profile (*Figure 7E*, top). In contrast, for the model with the highest effective dimension (denoted by a square marker in *Figure 7D*), aside from the sensory input and decision variable profiles, richer response kernel profiles were exhibited (*Figure 7E*, bottom). In particular, there was a set of single neuron response kernels with peak amplitudes occurring between the pulse input onset and the choice onset (*Figure 7E*, bottom right). These response kernels cannot be explained by the combination of sensory input and decision variables. Instead, the existence of these response kernels signifies neural dynamics in extra dimensions beyond the subspaces spanned by the input and decision variable, the genuine neural dynamical signature of the existence of selection vector modulation (*Figure 6C*).

While single neuron response kernels are illustrative in highlighting the model difference, they lack explanatory power at the population level. Therefore, we employed the singular value decomposition method to extract the principal dynamical modes of response kernels at the population level (see Methods for details). We found that similar dynamical modes, including one persistent choice

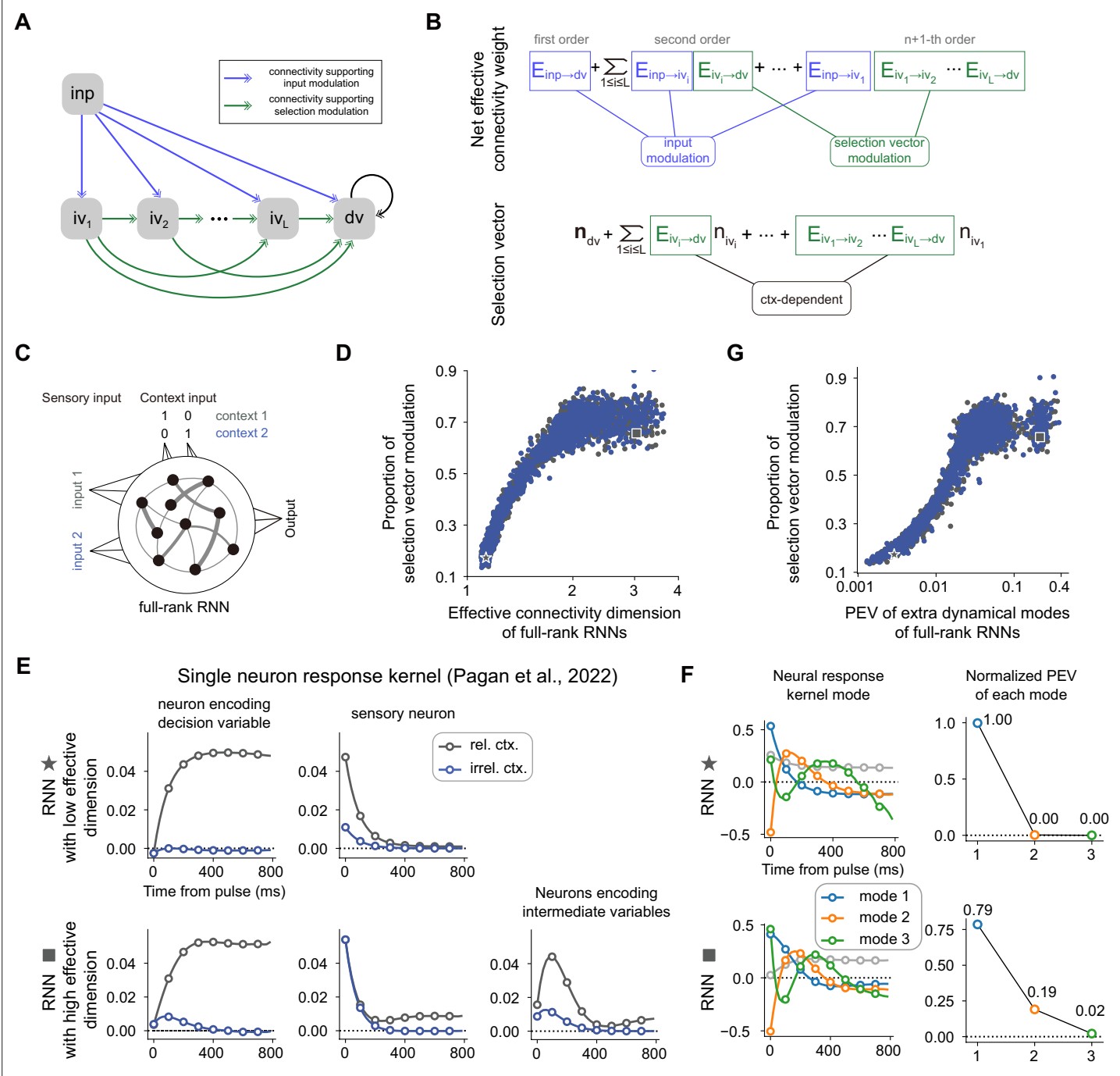

**Figure 7.** The correlation between the dimensionality of neural dynamics and the proportion of selection vector modulation is confirmed in vanilla recurrent neural networks (RNNs). (**A**) A general neural circuit model of context-dependent decision-making (CDM). In this model, there are multiple pathways capable of propagating the input information to the decision variable slot, of which the blue connections are susceptible to the input modulation while the green connections are susceptible to the selection vector modulation (see Methods for details). (**B**) The explicit formula of both the effective connection from the input variable to the decision variable and the effective selection vector for the model in (**A**). (**C**) The setting of vanilla RNNs trained to perform the CDM task. See Methods for more details. (**D**) Positive correlation between effective connectivity dimension and proportion of selection vector modulation. Given a trained RNN with matrix $J$, the effective connectivity dimension, defined by $\sum_{i=1}^{n} \sigma_i^2/\sigma_1^2$ where $\sigma_1 \geq \sigma_2 \geq \ldots \geq \sigma_n$ are singular values of $J$, is used to quantify the connectivity dimensionality. Spearman's rank correlation, $r$=0.919, p<1e-3, n=3892. The x-axis is displayed in log scale. (**E**) Single neuron response kernels for two example RNNs. The neuron response kernels were calculated using a regression method (*Pagan et al., 2022*; see Methods for details). For simplicity, only response kernels for input 1 are displayed. *Top*: Response kernels for two example neurons in the RNN with low effective dimension (indicated by a star marker in panel D). Two typical response kernels, including the

*Figure 7 continued on next page*

*Figure 7 continued*

decision variable profile (left) and the sensory input profile (right), are displayed. *Bottom*: Response kernels for three example neurons in the RNN with high effective dimension (indicated by a square marker in panel D). In addition to the decision variable profile (left) and sensory input profile (middle), there are neurons whose response kernels initially increase and then decrease (right). Gray lines, response kernels in context 1 (i.e. rel. ctx.). Blue lines, response kernels in context 2 (i.e. irrel. Ctx.). (**F**) Principal dynamical modes for response kernels in the population level extracted by singular value decomposition. *Left*: Shared dynamical modes, including one persistent choice mode (gray) and three transient modes (blue, orange, green) are identified across both RNNs. *Right*: For the $i$-th transient mode, the normalized percentage of explained variance (PEV) is given by $\sigma_i^2 / \sum_{j=1}^{39} \sigma_j^2$, where $\sigma_1 \geq \sigma_2 \geq \ldots \geq \sigma_{39}$ are singular values for each transient mode (see Methods for details). (**G**) Positive correlation between response-kernel-based index and proportion of selection vector modulation. For a given RNN, percentage of explained variance (PEV) of extra dynamical modes is defined as the accumulated normalized PEV of the second and subsequent transient dynamical modes (see Methods for details). Spearman's rank correlation, $r$=0.902, p<1e-3, n=3892. The x-axis is displayed in log scale.

The online version of this article includes the following figure supplement(s) for figure 7:

**Figure supplement 1.** Training vanilla recurrent neural networks (RNNs) with different regularization coefficients.

**Figure supplement 2.** Verification correlation results using vanilla recurrent neural networks (RNNs) trained with different hyper-parameter settings.

**Figure supplement 3.** Two recurrent neural networks (RNNs) with distinct modulation strategies produce the same neural activities.

**Figure supplement 4.** Artificially introducing redundant structure can disrupt the percentage of explained variance (PEV) of extra dynamical modes index.

mode (gray) and three transient modes (blue, orange, green), were shared across both the low and high effective dimension models (*Figure 7F*, left). The key difference between these two models lies in the percentage of explained variance (PEV) of the second transient mode (orange): while there is near-zero PEV in the low effective dimension model (*Figure 7F*, top right), there is substantial PEV in the high effective dimension model (*Figure 7F*, bottom right), consistent with the single neuron picture shown in *Figure 7E*. This result led us to use the PEV of extra dynamical modes (including the orange and green ones; see Methods for details) as a simple index to quantify the amount of selection vector modulation in these models. As expected, we found that the PEV of extra dynamical modes can serve as a reliable index reflecting the proportion of selection vector modulation in these models (*Figure 7G*, right panel and *Figure 7—figure supplement 1B and C*). Similar results for vanilla RNNs trained with different hyperparameter settings are displayed in *Figure 7—figure supplement 2*.

Together, we identified novel neural dynamical signatures of section vector modulation at both the single neuron and population level, suggesting the potential great utility of these neural dynamical signatures in distinguishing the contribution of selection vector modulation from input modulation in experimental data.

## Discussion

Using low-rank RNNs, we provided a rigorous theoretical framework linking network connectivity, neural dynamics, and selection mechanisms, and gained an in-depth algebraic and geometric understanding of both input and selection vector modulation mechanisms, and accordingly uncovered a previously unknown link between selection vector modulation and extra dimensions in neural state space. This gained understanding enabled us to generate novel predictions linking novel neural dynamic modes with the proportion of selection vector modulation, paving the way towards addressing the intricacy of neural variability across subjects in context-dependent computation.

### A pathway-based definition of selection vector modulation

In their seminal work, Mante, Sussillo, and their collaborators developed a numerical approach to compute the selection vector for trained RNN models (*Mante et al., 2013*). Based on this concept of selection vector, recently, Pagan et al. proposed a new theoretical framework to decompose the solution space of context-dependent decision-making, in which input modulation and selection vector modulation were explicitly defined (i.e. *Equation 1*). Here, taking the theoretical advantage of low-rank RNNs (*Mastrogiuseppe and Ostojic, 2018*; *Dubreuil et al., 2022*), we went beyond numerical reverse-engineering and provided a complementary pathway-based definition of both selection vector and selection vector modulation (i.e. *Equation 5*). This new definition gained us a novel geometric understanding of selection vector modulation, revealed a previously unknown link between extra dimensions and selection vector modulation (*Figure 6C*), and eventually provided us

with experimentally identifiable neural dynamical signature of selection vector modulation at both the single neuron and population levels (*Figure 7, E-G*).

## Individual neural variability in higher cognition

One hallmark of higher cognition is individual variability, as the same higher cognition problem can be solved equally well with different strategies. Therefore, studying the neural computations underlying individual variability is no doubt of great importance (*Hariri, 2009*; *Parasuraman and Jiang, 2012*; *Keung et al., 2020*; *Nelli et al., 2023*). Recent experimental advances enabled researchers to investigate this important issue in a systematic manner using delicate behavioral paradigms and large-scale recordings (*Pagan et al., 2022*). However, the computation underlying higher cognition is largely internal, requiring discovering novel neural activity patterns as internal indicators to differentiate distinct circuit mechanisms. In the example of context-dependent decision-making studied here, to differentiate selection vector modulation from input modulation, we found the PEV in extra dynamical modes is a reliable index for a wide variety of RNNs (*Figure 7D*, *Figure 7—figure supplement 2*). However, cautions have to be made here as we can conveniently construct counter-examples deviating from the picture depicted by this index. For instance, manually introducing additional dimensions that do not directly contribute to the computation can disrupt the index (*Figure 7—figure supplement 4A and B*). In the extreme scenario, we can construct two models with distinct circuit mechanisms (selection vector modulation and input modulation, respectively) but having the same neural activities (*Figure 7—figure supplement 3*), suggesting that any activity-based index alone would fail to make this differentiation. Then, why did the proposed index work for the trained vanilla RNNs shown in *Figure 7D*? Our lesion analysis suggests that the underlying reason is that the major variance in neural activity of vanilla RNNs learned through backpropagation is task-relevant (*Figure 7—figure supplement 4C*, see Methods for details). However, it is highly likely that task-irrelevant neural activity variance exists in higher brain regions, meaning the proposed index may not perform well in neural recordings. Therefore, our modeling work suggests that, to address the intricacy of individual variability of neural computations underlying higher cognition, integrative efforts incorporating not only large-scale neural activity recordings but also activity perturbations, neuronal connectivity knowledge, and computational modeling may be inevitably required.

## Beyond context-dependent decision-making

While we mainly focused on context-dependent decision-making tasks in this study, the issue of whether input or selection vector modulation prevails is not limited to the domain of decision-making. For instance, recent work by *Chen et al., 2024* demonstrated that during sequence working memory control (*Botvinick and Watanabe, 2007*; *Xie et al., 2022*), sensory inputs presented at different ordinal ranks first entered into a common sensory subspace and then were routed to the corresponding rank-specific working memory subspaces in monkey frontal cortex. Here, similar to the decision-making case (*Figure 1C*), the same issue arises: where is the input information selected by the context (here, the ordinal rank)? Can the presence of a common sensory subspace (similar to the presence of location information in both relevant and irrelevant contexts in *Figure 1C*) preclude the input modulation? The pathway-based understanding of input and selection vector modulation gained from CDM in this study may be transferable to address these similar issues.

## The role of transient dynamics in extra dimensions in context-dependent computation

In this study, we linked the selection vector modulation with transient dynamics (*Aoi et al., 2020*; *Soldado-Magraner et al., 2023*) in extra dimensions. While the transient dynamics in extra dimensions are not necessary in context-dependent decision-making here (*Dubreuil et al., 2022*; *Figure 2*), more complex context-dependent computation may require its presence. For example, recent work by *Tian et al., 2024* found that transient dynamics in extra subspaces is required to perform the switch operation (i.e. exchanging information in subspace 1 with information in subspace 2). Understanding how transient dynamics in extra dimensions contribute to complex context-dependent computation warrants further systematic investigation.

In summary, through low-rank neural network modeling, our work provided a parsimonious mechanistic account for how information can be selected along different pathways, making significant

contributions towards understanding the intriguing selection mechanisms in context-dependent computation.

# Materials and methods

## Key resources table

| Reagent type (species) or resource | Designation | Source or reference | Identifiers | Additional information |
| --- | --- | --- | --- | --- |
| Software, algorithm | Python | https://www.python.org/ | RRID:SCR_008394 | version: 3.9 |

## The general form of RNNs

We investigated networks of $N$ neurons with $S$ input channels, described by the following temporal evolution equation

$$\tau \frac{dx_i(t)}{dt} = -x_i(t) + \sum_{j=1}^{N} J_{ij}\phi(x_j(t)) + \sum_{s=1}^{S} I_{si}u_s(t) + \epsilon_i(t).$$  (6)

In this equation, $x_i(t)$ represents the activation of neuron $i$ at time $t$, $\tau$ denotes the characteristic time constant of a single neuron, and $\phi$ is a nonlinear activation function. Unless otherwise specified, we use the tanh function as the activation function. The coefficient $J_{ij}$ represents the connectivity weight from neuron $j$ to neuron $i$. The input $u_s(t)$ corresponds to the $s$-th input channel at time $t$, with feedforward weight $I_{si}$ to neuron $i$, and $\epsilon_i(t)$ represents white noise at time $t$. The network's output is obtained from the neuron's activity $\phi(x)$ through a linear projection:

$$z(t) = \frac{1}{N}\sum_{i=1}^{N} w_i\phi(x_i).$$  (7)

The connectivity matrix $J$, specified as $J = \{J_{ij}\}, i = 1, \ldots, N, j = 1, \ldots, N$, can be a low-rank or a full-rank matrix. In the low-rank case, $J$ is restricted to a low-rank matrix $\frac{1}{N}\sum_{r=1}^{R} m_r n_r^T$, in which $m_r$ is the $r$-th *output vector*, and $n_r$ is the $r$-th *input-selection vector*, with each element of $m_r$ and $n_r$ considered an independent parameter (**Dubreuil et al., 2022**). In this paper, we set the time constant $\tau$ to be 100 ms, and use Euler's method to discretize the evolution equation with a time step $\Delta t = 20ms$.

## Task setting

We modeled the click-version CDM task recently investigated by **Pagan et al., 2022**. The task involves four input channels $u_1(t)$, $u_2(t)$, $u_1^{ctx}(t)$, and $u_2^{ctx}(t)$, where $u_1(t)$ and $u_2(t)$ are stimulus inputs and $u_1^{ctx}(t)$ and $u_2^{ctx}(t)$ are context inputs. Initially, there is a fixation period lasting for $T_{fix} = 200ms$. This is followed by a stimulus period of $T_{sti} = 800ms$ and then a decision period of $T_{decision} = 20ms$.

For trial $k$, at each time step, the total number of pulse inputs (#pulse) is sampled from a Poisson distribution with a mean value of $40\Delta t = 0.8$. Each pulse has two properties: location and frequency. Location can be either left or right, and frequency can be either high or low. We randomly sample a pulse to be right with probability $p^k$ (hence left with probability $1 - p^k$) and to be high with probability $p_{high}^k$ (hence low with probability $1 - p_{high}^k$). The values of $p^k$ and $p_{high}^k$ are independently chosen from the set {39/40, 35/40, 25/40, 15/40, 5/40, 1/40}. The stimulus strength for the location input at trial $k$ is defined as $2 \times p^k - 1$, and for the frequency input, it is defined as $2 \times p_{high}^k - 1$. The input $u_1(t)$ represents location evidence, calculated as 0.1 times the difference between the number of right pulses and the number of left pulses (#right-#left) at time step $t$. The input $u_2(t)$ represents frequency evidence, calculated as 0.1 times the difference between the number of high-frequency pulses and the number of low-frequency pulses (#high-#low) at time step $t$.

The context is randomly chosen to be either the location context or the frequency context. In the location context, $u_1^{ctx} = 1$ and $u_2^{ctx} = 0$ throughout the entire period. The target output $z_k$ is defined as the sign of the location stimulus strength (1 if $p^k > 0.5$, otherwise –1). Thus, in this context, the target output $z_k$ is independent of the frequency stimulus input. Conversely, in the frequency context, $u_1^{ctx} = 0$ and $u_2^{ctx} = 1$. The target output $z_k$ is defined as the sign of the frequency stimulus strength (1

if $p_{high}^k > 0.5$, otherwise –1). Thus, in this context, the target output is independent of the location stimulus input.

## Linearized dynamical system analysis (Figure 1 and Figure 7)

To uncover the computational mechanism enabling each RNN to perform context-dependent evidence accumulation, we utilize linearized dynamical system analysis to 'open the black box' (*Mante et al., 2013*). The dynamical evolution of an RNN in context $c$ is given by:

$$\tau \frac{dx}{dt} = -x + Jr + I_1 u_1 + I_2 u_2 + I_c^{ctx},$$ (8)

where $r = \phi(x)$ is the neuron activity. First, we identify the slow point of each RNN in each context using an optimization method (*Mante et al., 2013*). Let $x$ be the discovered slow point, i.e., $x \approx Jr^* + I_c^{ctx}$ where $r = \phi(x^*)$. We define a diagonal matrix $G = \text{diag}\left(\phi'(x^*)\right)$, with the $i$-th diagonal element representing the sensitivity of the $i$-th neuron at the slow point. Near the slow point, we have $\Delta r = r - r^* \approx G\phi'(\Delta x)$. Then, we can derive:

$$\tau \frac{d\Delta r}{dt} \approx \tau G \frac{d\Delta x}{dt} = -\Delta r + GJ\Delta r + GI_1 u_1 + GI_2 u_2.$$ (9)

Thus, the dynamics of neuron activity around the slow point can be approximated by a linear dynamical system:

$$\tau \frac{d\Delta r}{dt} = M\Delta r + \widetilde{I}_1 u_1 + \widetilde{I}_2 u_2,$$ (10)

$$M = -E + GJ,$$ (11)

where $E$ is the identity matrix, $M$ denotes the state transition matrix of the dynamical system, and $\widetilde{I}_r = GI_r, r = 1, 2$ are stimulus *input representation directions*. Similar to previous work (*Mante et al., 2013*), we find that for every network, the linear dynamical system near the slow point in each context is roughly a linear attractor. Specifically, the transition matrix $M$ has a single eigenvalue close to zero, while all other eigenvalues have negative real parts. The right eigenvector of $M$ associated with the eigenvalue close to zero defines the stable direction of the dynamical system, forming the line attractor $\rho$(unit norm). The left eigenvector of $M$ associated with that eigenvalue defines the direction of the selection vector $s$. The norm of selection vector $s$ is chosen such that $s \cdot \rho = 1$. Previous work (*Mante et al., 2013*) has shown that a perturbation $\Delta r_0$ from the line attractor will eventually converge to the line attractor, with the distance from the starting point being $s \cdot \Delta r_0$.

Based on linearized dynamical systems analysis, Pagan et al. recently defined **input modulation** and **selection vector modulation** (*Pagan et al., 2022*) as:

$$mod_{inp} = \Delta GI \cdot \bar{s},$$ (12)

$$mod_{sel} = \bar{G}I \cdot \Delta s.$$ (13)

Specifically, the input modulation and selection vector modulation for stimulus input 1 are defined as $(s^{ctx_1} + s^{ctx_2})/2 \cdot \left(\widetilde{I}_1^{ctx_1} - \widetilde{I}_1^{ctx_2}\right)$ and $(s^{ctx_1} - s^{ctx_2}) \cdot \left(\widetilde{I}_1^{ctx_1} + \widetilde{I}_1^{ctx_2}\right)/2$, respectively. Similarly, the input modulation and selection vector modulation for stimulus input 2 are defined as $(s^{ctx_1} + s^{ctx_2})/2 \cdot \left(\widetilde{I}_2^{ctx_2} - \widetilde{I}_2^{ctx_1}\right)$ and $(s^{ctx_2} - s^{ctx_1}) \cdot \left(\widetilde{I}_2^{ctx_1} + \widetilde{I}_2^{ctx_2}\right)/2$, respectively. Proportion for selection vector modulation (*Figure 7D and G*) is defined as $\frac{mod_{sel}}{mod_{inp} + mod_{sel}}$.

## Training of rank-1 RNNs using backpropagation (Figure 2)

The rank-1 RNNs in *Figure 2* are trained using backpropagation-through-time with the PyTorch framework. These networks are trained to minimize a loss function defined as:

$$L = \sum_{k,t} M_t(\hat{Z}_{k,t} - Z_k) + L_{reg}.$$ (14)

Here, $z_k$ is the target output, $z_{k,t}$ is the network output, and the indices $t$ and $k$ represent time and trial, respectively. $M_t$ is a temporal mask with value $\{0, 1\}$, where $M_t$ is 1 only during the decision period. $L_{reg}$ is the L2 regularization loss. For full-rank RNNs (**Figure 7**), $L_{reg} = w_{reg} \sum_{ij} J_{ij}^2$ and for low-rank RNNs, $L_{reg} = w_{reg} \sum_r \left( \|m_r\|_2^2 + \|n_r\|_2^2 \right)$. The loss function is minimized by computing gradients with respect to all trainable parameters. We use the Adam optimizer in PyTorch with the decay rates for the first and second moments set to 0.9 and 0.99, respectively, and a learning rate of $10^{-3}$. Each RNN is trained for 5000 steps with a batch size of 256 trials.

For **Figure 2**, we trained 100 RNNs of $N = 512$ neurons with the same training hyperparameters. The rank of the connectivity matrix is constrained to be rank-1, represented as $J = \frac{1}{N} m_{dv} n_{dv}^T$. We trained the elements of the input vectors $I_1, I_2, I_1^{ctx}, I_2^{ctx}$, connectivity vector $m_{dv}, n_{dv}$, and the readout vector $w$. All trainable parameters were initialized with random independent Gaussian weights with a mean of 0 and a variance of $1/N^2$. The regularization coefficient $w_{reg}$ is set to $10^{-4}$.

## Trial-averaged analyses (Figures 2F and 3F)

For the trial-averaged analyses shown in **Figures 2f and 3f**, we followed a procedure similar to **Mante et al., 2013**. Specifically, at each time point $t$ and for each neuron $i$, we fit the following linear regression model to characterize how different task variables contribute to that neuron's activity:

$$r_{i,t}\left(k\right) = \beta_{choice;i,t} choice\left(k\right) + \beta_{inp_1;i,t} u_{1,k} + \beta_{inp_2;i,t} u_{2,k} + \beta_{context;i,t} context\left(k\right) + \beta_{0;i,t} \tag{15}$$

Here, $choice\left(k\right)$, $u_{1,k}$, $u_{2,k}$, and $context\left(k\right)$ represent the values of the corresponding variables on trial k. Next, for each regressor (choice, input 1, input 2, context), we pooled the resulting regression coefficients across neurons to get the time-vary regression vectors $\boldsymbol{\beta}_{choice;t}, \boldsymbol{\beta}_{inp_1;t}, \boldsymbol{\beta}_{inp_2;t}, \boldsymbol{\beta}_{context;t}$. For regressor $v$, we identified the time $t_v^{max}$ when the regression vector $\beta_{v;t}$ has maximum norm, and got the time-independent regression vectors:

$$t_v^{max} = \mathrm{argmax}_t \left\| \boldsymbol{\beta}_{v;t} \right\|_2, \tag{16}$$

$$\beta_v = \beta_{v;t_v^{max}} \tag{17}$$

Next, we assembled these vectors into a matrix $B = \left[\boldsymbol{\beta}_{choice} \boldsymbol{\beta}_{inp_1} \boldsymbol{\beta}_{inp_2} \boldsymbol{\beta}_{context}\right]$ and used QR decomposition to get the orthogonalized regression basis $\left[\boldsymbol{\beta}_{choice}^{\perp}, \boldsymbol{\beta}_{inp_1}^{\perp}, \boldsymbol{\beta}_{inp_2}^{\perp}, \boldsymbol{\beta}_{context}^{\perp}\right]$. Finally, we averaged neuronal activity across trials sharing the same condition (choice, context, input 1), and then projected this average activity onto the choice and input 1 axes. This process resulted in the trial-averaged population dynamics illustrated in **Figures 2F and 3F**.

## Proof of no selection vector modulation in rank-1 RNNs (Figure 2)

The transition matrix of neuron activity in the rank-1 RNN is given by:

$$M = -E + \frac{1}{N} G m_{dv} n_{dv}. \tag{18}$$

Multiplying $M^T$ on the right by $\boldsymbol{n}_{dv}$, we obtain

$$M^T \boldsymbol{n}_{dv} = -\boldsymbol{n}_{dv} + \left\langle \widetilde{\boldsymbol{m}}_{dv}, \boldsymbol{n}_{dv} \right\rangle \boldsymbol{n}_{dv} \approx 0, \tag{19}$$

where the $\langle \cdot, \cdot \rangle$ symbol is defined as $\langle \boldsymbol{a}, \boldsymbol{b} \rangle = \frac{1}{N} \sum_{i=1}^{N} a_i b_i$ for two vectors of length-$N$. The requirement for the linear attractor approximation, $\left\langle \widetilde{\boldsymbol{m}}_{dv}, \boldsymbol{n}_{dv} \right\rangle \approx 1$, is met by all our trained and handcrafted RNNs (**Figure 4—figure supplement 1**). This demonstrates that $n_{dv}$ is the left eigenvector of the transition matrix, and hence $\frac{1}{N} \left\| \widetilde{m}_{dv} \right\|_2 n_{dv}$ is the selection vector in each context. Therefore, the direction of the selection vector is invariant across different contexts for the rank-1 model, indicating no selection vector modulation and this is consistent with the training results shown in **Figure 2**.

## Handcrafting rank-3 RNNs with pure selection vector modulation (Figure 3)

First, we provide the implementation details of the rank-3 RNNs used in *Figure 3*. The network consists of 30,000 neurons divided into three populations, each with 10,000 neurons. The first population (neurons 1–10,000) receives stimulus input, accounting for both the information flow from the stimulus input to the intermediate variables and the recurrent connection of the decision variable. The second (neurons 10,001–20,000) and third populations (neurons 20,001–30,000) handle the information flow from the intermediate variables to the decision variable and are modulated by the context signal. To achieve this, we generate three Gaussian random matrices $(M^{(1)}, M^{(2)}, M^{(3)})$ of shape 10,000×3. Let $M^{(p)}_{:,r}$ denote the $r$-th column of matrix $M^{(p)}$. The stimulus input $I_1$ is given by the concatenation of three length-10,000 vectors $\left[M^{(1)}_{:,1}; 0; 0\right]$, 0 where denotes a length-10,000 zero vector. The stimulus input $I_2$ is given by $\left[M^{(1)}_{:,2}; 0; 0\right]$. The context input $I^{ctx}_1$ is given by $\left[0; M^{(2)}_{:,1}; 0\right]$. The context input $I^{ctx}_2$ is given by $\left[0; 0; M^{(3)}_{:,1}\right]$. The connectivity vectors $m_{iv1}, m_{iv2}$ and $m_{dv}$ are given by $\left[0; M^{(2)}_{:,2}; M^{(3)}_{:,2}\right]$, $\left[0; M^{(2)}_{:,3}; M^{(3)}_{:,3}\right]$ and $\left[M^{(1)}_{:,3}; 0; 0\right]$, respectively. The input-selection vectors $n_{iv1}, n_{iv2}$, and $n_{dv}$ are given by $\left[10M^{(1)}_{:,1}; 0; 0\right]$, $\left[10M^{(1)}_{:,2}; 0; 0\right]$, and $\left[3M^{(1)}_{:,3}; -gM^{(2)}_{:,2} + M^{(2)}_{:,3}; M^{(3)}_{:,2} - gM^{(3)}_{:,3}\right]$, respectively, where $g = \int^{+\infty}_{-\infty} \frac{1}{\sqrt{2\pi}} e^{-x^2/2} \phi'(x)\, dx$ represents the average gain of the second population in context 1 or the third population in context 2. The readout vector $w$ is given by $\left[4M^{(1)}_{:,3}; 0; 0\right]$. We generate 100 RNNs based on this method to ensure that the conclusions in *Figure 3* do not depend on the specific realization of random matrices. Linearized dynamical system analysis reveals that all these RNNs perform flexible computation through pure selection vector modulation. Please see the section '**The construction of rank-3 RNN models**' for a mean-field-theory-based understanding.

## Pathway-based information flow graph analysis of low-rank RNNs (Figure 4)

The dynamical evolution equation for low-rank RNN with $R$ ranks and $S$ input channels in context $c$ is given by

$$\frac{dx}{dt} = -x + \frac{1}{N}\sum_{r=1}^{R} m_r n_r^T \phi(x) + \sum_{s=1}^{S} I_s u_s + I_c^{ctx} \qquad (20)$$

Assuming $x(0) = I_c^{ctx}$ at $t = 0$, the dynamics of $x(t) - x(0)$ are always constrained in the subspace spanned by $\{m_r, r = 1, \ldots, R\}$ and $\{I_s, s = 1, \ldots, S\}$. Therefore, $x(t)$ can be expressed as a linear combination of these vectors: $x(t) = I_c^{ctx} + \sum_{r=1}^{R} k_r(t) m_r + \sum_{s=1}^{S} k_{p_s}(t) I_s$, leading to the following evolving dynamics of task variables:

$$\tau \frac{dk_{p_s}(t)}{dt} = -k_{p_s}(t) + u_s(t), \qquad (21)$$

$$\tau \frac{dk_r(t)}{dt} = -k_r(t) + \frac{1}{N} n_r^T \phi\left(I_c^{ctx} + \sum_{j=1}^{R} k_j(t) m_j + \sum_{s=1}^{S} k_{p_s}(t) I_s\right), \qquad (22)$$

where $k_{p_s}(t)$ denotes activation along the $s$-th input vector, termed the input task variable and $k_r(t)$ denotes activation along the $j$-th output vector, termed the internal task variable.

## The rank-1 RNN case

Therefore, for rank-1 RNNs, the latent dynamics of decision variable (internal task variable associated with $m_{dv}$) in context $c$ is given by

$$\tau \frac{dk_{dv}(t)}{dt} = -k_{dv}(t) + \frac{1}{N} n_{dv}^T \phi \left( I_c^{ctx} + k_{dv}(t) m_{dv} + \sum_{s=1}^{2} k_{p_s}(t) I_s \right). \tag{23}$$

Similar to the linearized dynamical systems analysis introduced earlier, we linearized *Equation 15* around $I_c^{ctx}$, obtaining the following linearized equation:

$$\tau \frac{dk_{dv}(t)}{dt} = -k_{dv}(t) + \frac{1}{N} \left( n_{dv}^T \phi \left( I_c^{ctx} \right) + n_{dv}^T \widetilde{m}_{dv}^{ctx_c} k_{dv}(t) + \sum_{s=1}^{2} n_{dv}^T \widetilde{I}_s^{ctx_c} k_{p_s}(t) \right), \tag{24}$$

where $\widetilde{m}_{dv}^{ctx_c} = G_c m_{dv}$ (termed as the decision variable representation direction) and $\widetilde{I}_s^{ctx_c} = G_c I_s$ (termed as the input representation direction), with $G_c$ equal to diag $\left( \phi'\left( I_c^{ctx} \right) \right)$. By denoting $\frac{1}{N} n_{dv}^T \widetilde{m}_{dv}^{ctx_c}$ and $\frac{1}{N} n_{dv}^T \widetilde{I}_s^{ctx_c}$ as $E_{dv \to dv}^{ctx_c}$ and $E_{p_s \to dv}^{ctx_c}$, respectively, together with the fact that $\frac{1}{N} n_{dv}^T \phi(I_c)$ is close to zero for all trained rank-1 RNNs, we obtain

$$\tau \frac{dk_{dv}(t)}{dt} = -k_{dv}(t) + E_{dv \to dv} \kappa_{dv}(t) + E_{p_1 \to dv}^{ctx_c} k_{p_1}(t) + E_{p_2 \to dv}^{ctx_c} k_{p_2}(t), \tag{25}$$

which is *Equation 3* in the main text.

## The rank-3 RNN case

Using a similar method, we can uncover the latent dynamics of rank-3 RNNs, as shown in *Figure 5*. Note that the rank-3 RNN in *Figure 3* is a special case of this more general form. The latent dynamics for internal task variables in context $c$ can be written as:

$$\tau \frac{dk_{iv_s}}{dt} = -k_{iv_s}(t) + \frac{1}{N} n_{iv_s}^T \phi \left( I_c^{ctx} + k_{dv} m_{dv} + \sum_{s=1}^{2} k_{iv_s} m_{iv_s} + \sum_{s=1}^{2} k_{inp_s} I_s \right), \tag{26}$$

$$\tau \frac{dk_{dv}}{dt} = -k_{dv}(t) + \frac{1}{N} n_{dv}^T \phi \left( I_c^{ctx} + k_{dv} m_{dv} + \sum_{s=1}^{2} k_{iv_s} m_{iv_s} + \sum_{s=1}^{2} k_{inp_s} I_s \right). \tag{27}$$

By applying the same first-order Taylor expansion, we obtain the following equations:

$$\tau \frac{dk_{iv_s}}{dt} = -k_{iv_s} + \frac{1}{N} n_{iv_s}^T \left( \widetilde{m}_{dv}^{ctx_c} k_{dv} + \sum_{s=1}^{2} \widetilde{m}_{iv_s}^{ctx_c} k_{iv_s} + \sum_{s'=1}^{2} \widetilde{I}_{s'}^{ctx_c} k_{inp_{s'}} \right), \tag{28}$$

$$\tau \frac{dk_{dv}}{dt} = -k_{dv} + \frac{1}{N} n_{dv}^T \left( \widetilde{m}_{dv}^{ctx_c} k_{dv} + \sum_{s=1}^{2} \widetilde{m}_{iv_s}^{ctx_c} k_{iv_s} + \sum_{s=1}^{2} \widetilde{I}_s^{ctx_c} k_{inp_s} \right). \tag{29}$$

We consider the case in which the intermediate variables ($k_{iv_s}, s = 1, 2$, internal task variables associated with $m_{iv_s}, s = 1, 2$, respectively) only receive information from the corresponding stimulus input, and the effective coupling of the recurrent connection is 1 in both contexts. Specifically, we assume:

$$\left\langle \widetilde{I}_{iv_{s'}}^{ctx_c}, n_{iv_s} \right\rangle = 0, s \neq s', c = 1, 2, \tag{30}$$

$$\left\langle \widetilde{m}_{iv_{s'}}^{ctx_c}, n_{iv_s} \right\rangle = 0, s, s' \in \{1, 2\}, c = 1, 2, \tag{31}$$

$$\left\langle \widetilde{m}_{dv}^{ctx_c}, n_{dv} \right\rangle = 1, c = 1, 2. \tag{32}$$

Our construction methods for rank-3 RNNs in *Figures 3 and 5* guarantee these conditions when the network is large enough (for example, $N = 30,000$ in our setting). Under these conditions, *Equations 20 and 21* can be simplified to:

$$\tau \frac{dk_{iv_s}}{dt} = -k_{iv_s} + E_{inp_s \to iv_s}^{ctx_c} k_{inp_s}, \tag{33}$$

$$\tau \frac{dk_{dv}}{dt} = E_{inp_1 \to dv}^{ctx_c} k_{inp_1} + E_{inp_2 \to dv}^{ctx_c} k_{inp_2} + E_{iv_1 \to dv}^{ctx_c} k_{iv_1} + E_{iv_2 \to dv}^{ctx_c} k_{iv_2}. \tag{34}$$

Suppose at $t = 0$, the network receives a pulse from input 1 with size $A_1$ and a pulse from input 2 with size $A_2$, which correspond to $u_1(t) = A_1 \tau \delta(t)$, $u_2(t) = A_2 \tau \delta(t)$. Under this condition, the expression for $k_{dv}$ is given by

$$k_{dv}(t) = \sum_{s=1}^{2} A_s \left( E_{inp_s \to dv}^{ctx_c} \left( 1 - e^{-t/\tau} \right) + E_{inp_s \to iv_s}^{ctx_c} E_{iv_s \to dv}^{ctx_c} \left( 1 - e^{-t/\tau} - \frac{t}{\tau} e^{-t/\tau} \right) \right). \tag{35}$$

From **Equation 27**, as $t \to \infty$, $k_{dv}$ will converge to $\sum_{s=1}^{2} A_s \left( E_{inp_s \to dv}^{ctx_c} + E_{inp_s \to iv_s}^{ctx_c} E_{iv_s \to dv}^{ctx_c} \right)$, providing a theoretical basis for the pathway-based information flow formula presented in **Figures 4 and 5**.

## Building rank-3 RNNs with both input and selection vector modulations (Figure 5)

### Understanding low-rank RNNs in the mean-field limit

The dynamics of task variables in low-rank RNNs can be mathematically analyzed under the mean-field limit ($N \to +\infty$) when each neuron's connectivity component is randomly drawn from a multivariate Gaussian mixture model (GMM) (**Beiran et al., 2021**; **Dubreuil et al., 2022**). Specifically, we assume that, for the $i$-th neuron, the connectivity component vector $\left\{ I_s^{(i)}, s = 1, \ldots, S, \left( I_c^{ctx} \right)^{(i)}, c = 1, 2, m_r^{(i)}, r = 1, \ldots, R, n_r^{(i)}, r = 1, \ldots, R \right\}$ is drawn independently from a GMM with $P$ components. The weight for the $j$-th component is $\alpha_j$, and this component is modeled as a Gaussian distribution with mean zero and covariance matrix $\Sigma^{(j)}$. Let $\Sigma_{\Im}^{(j)}$ denote the upper-left $(S + R + 2) \times (S + R + 2)$ submatrix of $\Sigma^{(j)}$, which represents the covariance matrix of $\left\{ I_s^{(i)}, s = 1, \ldots, S, \left( I_c^{ctx} \right)^{(i)}, c = 1, 2, m_r^{(i)}, r = 1, \ldots, R \right\}$ within the $j$-th component. Let $\sigma_{ab}^{(p)}, a, b \in \{ I_s, s = 1, \ldots, S, I_c^{ctx}, c = 1, 2, m_r, r = 1, \ldots, R, n_r, r = 1, \ldots, R \}$ denote the covariance of $a^{(i)}$ and $b^{(i)}$, where the $i$-th neuron belongs to the $p$-th component.

Given these assumptions, under the mean-field limit ($N \to +\infty$), **Equation 14** can be expressed as

$$\tau \frac{dk_r(t)}{dt} \wedge -k_r(t) + \sum_{j=1}^{R} \sum_{p=1}^{P} \alpha_p \left\langle \phi' \right\rangle_p \sigma_{m_j n_r}^{(p)} k_j(t)$$
$$+ \sum_{s=1}^{S} \sum_{p=1}^{P} \alpha_p \left\langle \phi' \right\rangle_p \sigma_{I_s n_r}^{(p)} k_{inp_s} + \sum_{p=1}^{P} \alpha_p \left\langle \phi' \right\rangle_p \sigma_{I_c^{ctx} n_r}^{(p)}, \tag{36}$$

$$\left\langle \phi' \right\rangle_p = \int_{-\infty}^{\infty} \frac{1}{\sqrt{2\pi}} e^{\frac{-x^2}{2}} \phi'\left( \Delta_p x \right) dx, \tag{37}$$

$$\Delta_p^2 = \left[ k_{inp_1}, \ldots, k_{inp_S}, k_{ctx_1}, k_{ctx_2}, k_1, \ldots, k_R \right] \Sigma_{\Im}^{(p)} \left[ k_{inp_1}, \ldots, k_{inp_S}, k_{ctx_1}, k_{ctx_2}, k_1, \ldots, k_R \right]^T, \tag{38}$$

where $k_{ctx_i} = 1$ if $i = c$, otherwise $k_{ctx_i} = 0$. Under the condition of small task variables ($k_{p_s}, s = 1, \ldots, S$ and $k_r, r = 1, \ldots, R$), $\Delta_p^2$ is approximately equal to $\sigma_{I_c^{ctx}, I_c^{ctx}}^{(p)}$ and the quantities $\left\langle \phi' \right\rangle_p, p = 1, \ldots, P$ are determined solely by the covariance of the context $c$ signal input within each population. Clearly, the effective coupling from the input task variable $k_{p_s}$ to the internal task variable $k_r$ is given by $\sum_{p=1}^{P} \alpha_p \left\langle \phi' \right\rangle_p \sigma_{I_s n_r}^{(p)}$, and the effective coupling between the internal task variables $k_j$ and $k_r$ is given by $\sum_{p=1}^{P} \alpha_p \left\langle \phi' \right\rangle_p \sigma_{m_j n_r}^{(p)}$.

### Mean-field-theory-based model construction

Utilizing this theory, we can construct RNNs tailored to any given ratio of input modulation to selection vector modulation by properly setting the connectivity vectors ($\boldsymbol{I}_1, \boldsymbol{I}_2, \boldsymbol{I}_1^{ctx}, \boldsymbol{I}_2^{ctx}, \boldsymbol{m}_{iv1}, \boldsymbol{m}_{iv2}, \boldsymbol{m}_{dv}, \boldsymbol{n}_{iv1}, \boldsymbol{n}_{iv2}, and \boldsymbol{n}_{dv}$). The RNN we built consists of 30,000 neurons divided into three populations. The first population (neurons 1–10,000) receives the stimulus input, accounting for information flow from stimulus input to intermediate variables (the connection strength is controlled by $\beta$) and the recurrent connection of the decision variable. The second (neurons 10,001–20,000) and third populations (neurons 20,001–30,000) receive the stimulus input, accounting for the information flow from the stimulus input to the decision variable (the connection strength is controlled by $\alpha$), and the information flow from

the intermediate variables to the decision variable (the connection strength is controlled by $\eta$), and are modulated by contextual input. To achieve this, we generate three Gaussian random matrices $(M^{(1)}, M^{(2)}, M^{(3)})$ of shape $10{,}000 \times 3$, $10{,}000 \times 5$ and $10{,}000 \times 5$, respectively. Let $M^{(p)}_{:,r}$ denote the $r$-th column of matrix $M^{(p)}$. The stimulus input $I_1$ is given by the concatenation of three length-$10{,}000$ vectors $\left[M^{(1)}_{:,1}; M^{(2)}_{:,1}; M^{(3)}_{:,1}\right]$. The stimulus input $I_2$ is given by $\left[M^{(1)}_{:,2}; M^{(2)}_{:,2}; M^{(3)}_{:,2}\right]$. The context input $I^{ctx}_1$ is given by $\left[\mathbf{0}; M^{(2)}_{:,3}; \mathbf{0}\right]$. The context input $I^{ctx}_2$ is given by $\left[\mathbf{0}; \mathbf{0}; M^{(3)}_{:,3}\right]$. The connectivity vectors $\boldsymbol{m}_{iv1}, \boldsymbol{m}_{iv2}$, and $\boldsymbol{m}_{dv}$ are given by $\left[\mathbf{0}; M^{(2)}_{:,4}; M^{(3)}_{:,4}\right], \left[\mathbf{0}; M^{(2)}_{:,5}; M^{(3)}_{:,5}\right]$ and $\left[M^{(1)}_{:,3}; \mathbf{0}; \mathbf{0}\right]$, respectively. The input-selection vectors $\boldsymbol{n}_{iv_1}$ and $\boldsymbol{n}_{iv_2}$ are given by $\left[3\beta M^{(1)}_{:,1}; \mathbf{0}; \mathbf{0}\right], \left[3\beta M^{(1)}_{:,2}; \mathbf{0}; \mathbf{0}\right]$, respectively. $\boldsymbol{n}_{dv}$ is given by
$\left[3M^{(1)}_{:,3}; -\frac{3\alpha g}{1-g^2}M^{(2)}_{:,1} + \frac{3\alpha}{1-g^2}M^{(2)}_{:,2} - \frac{3\eta g}{1-g^2}M^{(2)}_{:,4} + \frac{3\eta}{1-g^2}M^{(2)}_{:,5}; \frac{3\alpha}{1-g^2}M^{(3)}_{:,1} - \frac{3\alpha g}{1-g^2}M^{(3)}_{:,2} + \frac{3\eta}{1-g^2}M^{(3)}_{:,4} - \frac{3\eta g}{1-g^2}M^{(3)}_{:,5}\right]$,
where $g = \int_{-\infty}^{+\infty} \frac{1}{\sqrt{2\pi}} e^{-x^2/2} \phi'(x)\, dx$ is the average gain of the second population in context 1 or third population in context 2. The readout vector $\boldsymbol{w}$ is given by $\left[4M^{(1)}_{:,3}; \mathbf{0}; \mathbf{0}\right]$.

For the network, in context 1, $\left\langle \phi' \right\rangle_p$ is only determined by the covariance of the context 1 signal. That is,

$$\left\langle \phi' \right\rangle_p = \int_{-\infty}^{\infty} \frac{1}{\sqrt{2\pi}} e^{\frac{-x^2}{2}} \phi'\left(\sqrt{\sigma^{(p)}_{I^{ctx}_1 I^{ctx}_1}} x\right) dx. \tag{39}$$

Our construction method guarantees that $\sigma^{(p)}_{I^{ctx}_1 I^{ctx}_1} = 0, 1, 0$ for $p = 1, 2, 3$, respectively. Hence, $\left\langle \phi' \right\rangle_p = 1, g, 1$ for $p = 1, 2, 3$, respectively. Then the effective coupling from input variable $k_{p_1}$ to $k_{iv_1}$ in context 1 is given by

$$E^{ctx_1}_{inp_1 \to iv_1} = \sum_{p=1}^{P} \frac{1}{3} \left\langle \phi' \right\rangle_p \sigma^{(p)}_{I_1 n_{iv_1}} = \frac{1}{3} \times 3 \times \beta = \beta. \tag{40}$$

Similarly, we can get that:

$$E^{ctx_1}_{inp_2 \to iv_2} = \beta, \tag{41}$$

$$E^{ctx_1}_{inp_1 \to dv} = \alpha, \quad E^{ctx_1}_{inp_2 \to dv} = 0, \tag{42}$$

$$E^{ctx_1}_{iv_1 \to dv} = \eta, \quad E^{ctx_1}_{iv_2 \to dv} = 0. \tag{43}$$

Other unmentioned effective couplings are zero. Similarly, the effective couplings in context 2 are given by

$$E^{ctx_2}_{inp_1 \to iv_2} = E^{ctx_2}_{inp_2 \to iv_2} = \beta, \tag{44}$$

$$E^{ctx_2}_{inp_1 \to dv} = 0, \quad E^{ctx_2}_{inp_2 \to dv} = \alpha, \tag{45}$$

$$E^{ctx_2}_{iv_1 \to dv} = 0, \quad E^{ctx_2}_{iv_2 \to dv} = \eta. \tag{46}$$

Thus, for each input, the input modulation is $\alpha$ and the selection vector modulation is $\beta \times \eta$. Therefore, any given ratio of input modulation to selection vector modulation can be achieved by varying the parameters $\alpha, \beta$, and $\eta$. The example in *Figure 3* with pure selection vector modulation is a special case with $\alpha = 0, \beta = \frac{10}{3}$, and $\eta = \left(1 - g^2\right)/3$.

## Pathway-based definition of selection vector (Figure 6)

Next, we will consider the rank-3 RNN with latent dynamics depicted in *Equations 18 and 19*. The input representation produced by a pulse input $u_s = \tau A_s \delta(t), s = 1, 2$ in a certain context at $t = 0$ is given by $A_1 \widetilde{\boldsymbol{I}}_1 + A_2 \widetilde{\boldsymbol{I}}_2$. We have proven that this pulse input will ultimately reach the $dv$ slot with a magnitude of $\sum_{s=1}^{2} A_s \left(E_{p_s \to dv} + E_{p_s \to iv_s} E_{iv_s \to dv}\right)$. This equation can be rewritten as

$$\frac{1}{N} \left( A_1 \widetilde{\boldsymbol{I}}_1 + A_2 \widetilde{\boldsymbol{I}}_2 \right) \cdot \left( \boldsymbol{n}_{dv} + E_{iv_1 \to dv} \boldsymbol{n}_{iv_1} + E_{iv_2 \to dv} \boldsymbol{n}_{iv_2} \right) \tag{47}$$

Therefore, we define $\boldsymbol{n}_{tol} = \boldsymbol{n}_{dv} + E_{iv_1 \to dv} \boldsymbol{n}_{iv_1} + E_{iv_2 \to dv} \boldsymbol{n}_{iv_2}$ as the pathway-based definition of the selection vector. Through calculations, we can prove that this pathway-based definition of the selection vector is equivalent to the classical definition based on linearized dynamical systems. In fact, the transition matrix of neuron activity in this rank-3 RNN is given by:

$$\begin{aligned} M = &-E + \frac{1}{N} G \left( \boldsymbol{m}_{iv_1} \boldsymbol{n}_{iv_1}^T + \boldsymbol{m}_{iv_2} \boldsymbol{n}_{iv_2}^T + \boldsymbol{m}_{dv} \boldsymbol{n}_{dv}^T \right) \\ &-E + \frac{1}{N} \left( \widetilde{\boldsymbol{m}}_{iv_1} \boldsymbol{n}_{iv_1}^T + \widetilde{\boldsymbol{m}}_{iv_2} \boldsymbol{n}_{iv_2}^T + \widetilde{\boldsymbol{m}}_{dv} \boldsymbol{n}_{dv}^T \right). \end{aligned} \tag{48}$$

Multiplying $M^T$ on the right by $\boldsymbol{n}_{tol}$, we obtain:

$$M^T \boldsymbol{n}_{tol} = 0. \tag{49}$$

Moreover, under the condition that the choice axis is invariant across contexts, i.e., $\widetilde{\boldsymbol{m}}_{dv}$ is invariant across contexts (*Mante et al., 2013*; *Pagan et al., 2022*),

$$\frac{\widetilde{\boldsymbol{m}}_{dv}}{\left\| \widetilde{\boldsymbol{m}}_{dv} \right\|_2} \cdot \left( \frac{1}{N} \left\| \widetilde{\boldsymbol{m}}_{dv} \right\|_2 \boldsymbol{n}_{tol} \right) = \left\langle \widetilde{\boldsymbol{m}}_{dv}, \boldsymbol{n}_{tol} \right\rangle = 1. \tag{50}$$

This demonstrates that $\frac{1}{N} \left\| \widetilde{\boldsymbol{m}}_{dv} \right\|_2 \boldsymbol{n}_{tol}$ is indeed the left eigenvector of the transition matrix as well as the classical selection vector of the linearized dynamical systems.

## The equivalence between two definitions of selection vector modulation (Figure 5)

Here, we use the rank-3 RNN and input 1 as an example to explain why there is an equivalence between our definition of selection vector modulation and the classical one (*Pagan et al., 2022*). In the previous section, we have proven that the input representation direction and selection vector are given by $\widetilde{\boldsymbol{I}}_1$ and $s = \frac{1}{N} \left\| \widetilde{\boldsymbol{m}}_{dv} \right\|_2 \left( \boldsymbol{n}_{dv} + E_{iv_1 \to dv} \boldsymbol{n}_{iv_1} + E_{iv_2 \to dv} \boldsymbol{n}_{iv_2} \right)$, respectively. According to the classical definition (*Pagan et al., 2022*), the input modulation and selection vector modulation are given by $mod_{inp} = \Delta \widetilde{\boldsymbol{I}}_1 \cdot \bar{s}$ and $mod_{sel} = \widetilde{\boldsymbol{I}}_1 \cdot \Delta s$, respectively. Since $\widetilde{\boldsymbol{m}}_{dv}, \boldsymbol{n}_{iv_1}, \boldsymbol{n}_{iv_2}$, and $\boldsymbol{n}_{dv}$ are invariant across different contexts, we have:

$$\bar{s} = \frac{1}{N} \left\| \widetilde{\boldsymbol{m}}_{dv} \right\|_2 \left( \boldsymbol{n}_{dv} + \bar{E}_{iv_1 \to dv} \boldsymbol{n}_{iv_1} + \bar{E}_{iv_2 \to dv} \boldsymbol{n}_{iv_2} \right), \tag{51}$$

$$\Delta s = \frac{1}{N} \left\| \widetilde{\boldsymbol{m}}_{dv} \right\|_2 \left( \Delta E_{iv_1 \to dv} \boldsymbol{n}_{iv_1} + \Delta E_{iv_2 \to dv} \boldsymbol{n}_{iv_2} \right). \tag{52}$$

Substituting these into *Equations 7 and 8* yields:

$$mod_{inp} = \left\| \widetilde{\boldsymbol{m}}_{dv} \right\|_2 \left( \Delta E_{inp_1 \to dv} + \Delta E_{inp_1 \to iv_1} \bar{E}_{iv_1 \to dv} \right), \tag{53}$$

$$mod_{sel} = \left\| \widetilde{\boldsymbol{m}}_{dv} \right\|_2 \left( \bar{E}_{inp_1 \to iv_1} \Delta E_{iv_1 \to dv} \right), \tag{54}$$

a result fully consistent with the pathway-based definition in *Equation 5* of the main text.

## Comparison between the pathway-based selection vector and the classical one (Figure 6)

We generate 1000 RNNs according to the procedure in *Figure 5C* (see Method '*Mean-field-theory-based model construction*' for details), with each RNN defined by parameters $\alpha$, $\beta$ and $\gamma$ independently sampled from a Uniform (0, 1) distribution. For each RNN, we computed the selection vector for the RNN in a given context (e.g. context 1 or 2) in two ways:

via linearized dynamical system analysis following *Mante et al., 2013*, producing the selection vector sv$^{classical}$ (classical in *Figure 6B*), using the theoretical derivation sv $= n_{dv} + \eta n_{iv_1}$ ('our's' in *Figure 6B*).

We repeated this process 1000 times and measured the cosine angle between these two selection vectors and plotted the resulting distribution for context 1 (gray) and context 2 (blue) in *Figure 6B*.

## Pathway-based analysis of higher order low-rank RNNs (Figure 7A and B)

In this section, we will consider RNNs with more intermediate variables (*Figure 7A*). Here, we consider only one stimulus modality for simplicity and use the same notation convention as in the previous sections. The network consists of one input variable $k_{inp}$, $L$ intermediate variables $k_{iv_i}, i = 1, \ldots, L$, and one decision variable $k_{dv}$. For simplicity, we let $k_{iv_0}$ be an alias for the input variable and $k_{iv_{L+1}}$ be an alias for the decision variable. We use the shorthand $E_{i \to j}$ and $E_{i \to dv}$ to denote the effective coupling $E_{iv_i \to iv_j}$ and $E_{iv_i \to dv}$, respectively. The dynamics of these task variables are given by

$$\tau \frac{dk_{inp}}{dt} = -k_{inp} + u, \tag{55}$$

$$\tau \frac{dk_{iv_i}}{dt} = -k_{iv_i} + \sum_{j=0}^{i-1} E_{j \to i} k_{iv_j}, i = 1, \ldots, L, \tag{56}$$

$$\tau \frac{dk_{dv}}{dt} = \sum_{j=0}^{L} E_{j \to L+1} k_{iv_j}. \tag{57}$$

Consider the case when, at time $t = 0$, the network receives a pulse input with unit size (i.e. $u(t) = \tau \delta(t)$). Then, we have

$$k_{inp} = e^{\frac{-t}{\tau}}, \tag{58}$$

$$k_{iv_i} = \sum_{j=1}^{i} \frac{K_j^i}{j!} e^{\frac{-t}{\tau}} \left(\frac{t}{\tau}\right)^j i = 0, \ldots, n, \tag{59}$$

$$k_{dv} = k_{iv_{L+1}} = \sum_{j=1}^{L+1} K_j^{L+1} \left(1 - \frac{\Gamma\left(j, \frac{t}{\tau}\right)}{(j-1)!}\right), \tag{60}$$

where $K_j^i = \sum 0 = a_1 < a_2 < \ldots < a_j < i E_{a_1 \to a_2} \cdot E_{a_2 \to a_3} \cdot \ldots \cdot E_{a_j \to i}$. In *Equation 52*, $\Gamma$ stands for the incomplete Gamma function and $\Gamma\left(j, \frac{t}{\tau}\right) = \int_{t/\tau}^{\infty} x^{j-1} e^{-x} dx$. These expressions tell us that, as time goes to infinity, all intermediate task variables $k_{iv_i}, i = 1, \ldots, L$ will decay to zero and the decision variable will converge to $\sum_{j=1}^{L+1} K_j^{L+1}$, which means that a pulse input of unit size will ultimately reach the $dv$ slot with a magnitude of $\sum_{j=1}^{L+1} K_j^{L+1}$. Therefore, we define the total effective coupling from the input variable to the decision variable in this higher-order graph as

$$E_{tol} = \sum_{j=1}^{L+1} \sum_{0=a_1 < a_2 < \ldots < a_j \le L} E_{a_1 \to a_2} \cdot E_{a_2 \to a_3} \cdot \ldots \cdot E_{a_j \to dv}. \tag{61}$$

The difference of the total effective coupling between relevant context and irrelevant context can be decomposed into:

$$\Delta E_{tol} = \sum_{j=1}^{L+1} \sum_{0=a_1 < a_2 < \ldots < a_j \le L} \Delta\left(E_{a_1 \to a_2}\right) \cdot \bar{E}_{a_2 \to a_3} \cdot \ldots \cdot \bar{E}_{a_j \to dv} +$$
$$\sum_{j=1}^{L+1} \sum_{0=a_1 < a_2 < \ldots < a_j \le L} \bar{E}_{a_1 \to a_2} \cdot \Delta\left(E_{a_2 \to a_3} \cdot \ldots \cdot E_{a_j \to dv}\right). \tag{62}$$

The first term, caused by changing a stimulus input representation, is defined as the input modulation. The second term, the one without changing the stimulus input representation, is defined as the selection vector modulation.

Using a similar method in rank-3 RNNs, the selection vector for RNNs of higher order is given by:

$$n_{dv} + \sum_{j=1}^{L} \sum_{0 < a_1 < \ldots < a_j \leq L} \left( E_{a_1 \to a_2} \cdot \ldots \cdot E_{a_j \to dv} \right) n_{iv_{a_1}}. \tag{63}$$

## Training of full-rank vanilla RNNs using backpropagation (Figure 7)

For *Figure 7*, we trained full-rank RNNs of $N = 128$ neurons. We trained the elements of the input vectors, the connectivity matrix, and the readout vector. We tested a large range of regularization coefficients ranging from 0 to 0.1. For each $r_{reg}$ chosen from the set {0, 0.001, 0.002, 0.003, 0.004, 0.005, 0.006, 0.007, 0.008, 0.009, 0.01, 0.02, 0.03, 0.04, 0.05, 0.06, 0.07, 0.08, 0.09, 0.1}, we trained 100 full-rank RNNs. A larger $w_{reg}$ value results in a trained connectivity matrix $J$ with lower rank, making the network more similar to a rank-1 RNN. All trainable parameters were initialized with random independent Gaussian weights with a mean of 0 and variance of $1/N^2$. Only trained RNNs with their largest eigenvalue of the activity transition matrix falling within the –0.05–0.05 range in both contexts were selected for subsequent analysis.

To ensure that the conclusions drawn from *Figure 7* are robust and not dependent on specific hyperparameter settings, we conducted similar experiments under different model hyperparameter settings. First, we trained RNNs with the softplus activation function and regularization coefficients chosen from {0.1, 0.09, 0.08, 0.07, 0.06, 0.05, 0.04, 0.03, 0.02, 0.01, 0.008, 0.004, 0.002, 0.001} (*Figure 7—figure supplement 2A*). Unlike tanh, softplus does not saturate on the positive end. Additionally, we tested the initialization of trainable parameters with a variance of $1/N$ (*Figure 7—figure supplement 2B*). Together, these experiments confirmed that the main results do not depend on the specific model hyperparameter settings.

## Estimating matrix dimension and extra dimension (*Figure 7*)
### Effective dimension of connectivity matrix (Figure 7D)
Let $\sigma_1 \geq \sigma_2 \geq \ldots \geq \sigma_n$ be the singular values of the connectivity matrix. The matrix's rank is the number of nonzero singular values. However, rank alone can overlook differences in how quickly those singular values decay. To capture this, we define the effective dimension as its stable rank (*Sanyal et al., 2020*):

$$\text{edim}(J) = \sum_{i=1}^{n} \frac{\sigma_i^2}{\sigma_1^2}. \tag{64}$$

Each term lies between 0 and 1, so the effective dimension satisfies $0 \leq \text{edim}(J) \leq rank(J)$. When all nonzero singular values are equal, $\text{edim}(J)$ equals the matrix rank. But if some singular values are much smaller than others, effective dimension will be closer to 1.

### Single neuron response kernel for pulse input (Figure 7E)
We apply pulse-based linear regression (*Pagan et al., 2022*) to assess how pulse input affects neuron activity. The activity of neuron $i$ in trial $k$ at time step $t$ is given by:

$$r_{i,t}(k) = \beta_{choice;i,t} choice(k) + \beta_{context;i,t} context(k) + \beta_{time;i,t} + \beta_{inp_1,ctx_c;i} * u_{1,k} + \beta_{inp_2,ctx_c;i} * u_{2,k}, \tag{65}$$

where $choice(k)$ is the RNN's choice on trial $k$ defined as the sign of its output during the decision period, $context(k)$ is the context of trial $k$ (1 for context 1 and 0 for context 2), $u_{i,k}$ indicates signed input $i$ evidence as defined previously in Methods. The first three regression coefficients, $\beta_{choice;i,t}$, $\beta_{context;i,t}$ and $\beta_{time;i,t}$, capture the influence of choice, context, and time on the neuron's activity at each time step, each being a 40-dimension vector. The remaining coefficient, $\beta_{inp_1,ctx_1;i}$, $\beta_{inp_1,ctx_2;i}$, $\beta_{inp_2,ctx_1;i}$, and $\beta_{inp_2,ctx_2;i}$, reflect the impact of pulse input on the neuron activity in a specific context, each also a 40-dimension vector. The asterisk (*) indicates the convolution operation between the response kernel and input evidence, described by:

$$\left(\beta_{inp_1,ctx_c;i} * u_{1,k}\right)(t) = \sum_{s=1}^{t} u_{1,k}(s)\, \beta_{inp_1,ctx_c;i}(t-s). \tag{66}$$

Therefore, there are a total of 280 regression coefficients for each neuron. We obtained these coefficients using ridge regression with 1000 trials for each RNN. The coefficient $\beta_{inp_i,ctx_c;i}(t)$ is termed the single neuron response kernel for input $i$ in context $c$ (**Figure 7E**).

## Response kernel modes and normalized percentage of explained variance (PEV) (Figure 7F)

For each RNN, we construct a matrix $B$ with shape $N_t \times (2N)$ from neuron response kernels for input 1 (or input 2) across both contexts, where $N_t = 40$ represents time steps and $N$ is the number of neurons, with each column as a neuron's response kernel in a context. Then we apply singular value decomposition (SVD) to the matrix:

$$B = USV^T \tag{67}$$

where $S$ is a diagonal square matrix of size $N_t$, with diagonal element $\sigma_1 \geq \sigma_2 \geq \ldots \geq \sigma_{N_t}$ being the singular value of the matrix. The first column of U serves as a persistent mode (mode 0), while the second and following columns are transient modes (transient dynamical modes 1, 2, etc.). The normalized PEV of transient dynamical mode $r, r \geq 1$ is defined as:

$$PEV_{mode_r} = \frac{\sigma_{r+1}^2}{\sum_{i=2}^{40} \sigma_i^2}. \tag{68}$$

## PEV of extra dynamical modes

The PEV of extra dynamical modes for input 1 (or 2) is defined based on the normalized PEV of transient dynamical mode:

$$PEV_{ED}\left(inp_1\right) = \sum_{r=1}^{N_t-1} PEV_{mode_r}. \tag{69}$$

## The counterintuitive extreme example (Figure 7—figure supplement 3)

We can manually construct two models (RNN1 and RNN2) with distinct circuit mechanisms (input modulation versus selection vector modulation) but showing the same neural activities. Specifically, we generated three Gaussian random matrices ($M^{(1)}, M^{(2)}, M^{(3)}$) of shape $10{,}000 \times 5$, $10{,}000 \times 5$ and $10{,}000 \times 1$, respectively. Let $M_{:,r}^{(p)}$ denote the $r$-th column of matrix $M^{(p)}$. The two models have the same input vectors, output vectors, and input-selection vectors for intermediate task variables ($\boldsymbol{n}_{iv_1}$ and $\boldsymbol{n}_{iv_2}$), given by:

$$\boldsymbol{I}_1 = \left[M_{:,1}^{(1)}; M_{:,1}^{(2)}; \boldsymbol{0}\right], \boldsymbol{I}_2 = \left[M_{:,2}^{(1)}; M_{:,2}^{(2)}; \boldsymbol{0}\right] \tag{70}$$

$$\boldsymbol{I}_1^{ctx} = \left[M_{:,5}^{(1)}; \boldsymbol{0}; \boldsymbol{0}\right], \boldsymbol{I}_2^{ctx} = \left[\boldsymbol{0}; M_{:,5}^{(2)}; \boldsymbol{0}\right] \tag{71}$$

$$\boldsymbol{m}_{iv_1} = \left[M_{:,3}^{(1)}; M_{:,3}^{(2)}; \boldsymbol{0}\right], \boldsymbol{m}_{iv_2} = \left[M_{:,4}^{(1)}; M_{:,4}^{(2)}; \boldsymbol{0}\right], \boldsymbol{m}_{dv} = \left[\boldsymbol{0}; \boldsymbol{0}; M_{:,1}^{(3)}\right] \tag{72}$$

$$\boldsymbol{n}_{iv_1} = \left[\frac{3}{1+g}M_{:,1}^{(1)}; \frac{3}{1+g}M_{:,1}^{(2)}; \boldsymbol{0}\right], \boldsymbol{n}_{iv_2} = \left[\frac{-3g}{1-g^2}M_{:,1}^{(1)}; \frac{3}{1-g^2}M_{:,1}^{(2)}; \boldsymbol{0}\right], \tag{73}$$

where $g = \int_{-\infty}^{+\infty} \frac{1}{\sqrt{2\pi}} e^{-x^2/2}\phi'(x)\, dx$. The difference between these two RNNs lies in their input-selection vector $\boldsymbol{n}_{dv}$ for the decision variable. For RNN1, $\boldsymbol{n}_{dv}$ is given by

$$\boldsymbol{n}_{dv} = \left[\frac{3}{1+g}M_{:,4}^{(1)}; \frac{3}{1+g}M_{:,4}^{(2)}; \boldsymbol{0}\right] + \left[\frac{3}{1-g^2}M_{:,2}^{(1)}; \frac{-3g}{1-g^2}M_{:,2}^{(2)}; \boldsymbol{0}\right] + \left[\boldsymbol{0}; \boldsymbol{0}; 3M_{:,1}^{(3)}\right]. \tag{74}$$

For RNN2, $\boldsymbol{n}_{dv}$ is given by:

$$\boldsymbol{n}_{dv} = \left[\frac{-3g}{1-g^2}M^{(1)}_{:,3}; \frac{3}{1-g^2}M^{(2)}_{:,3}; \boldsymbol{0}\right] + \left[\frac{3}{1-g^2}M^{(1)}_{:,2}; \frac{-3g}{1-g^2}M^{(2)}_{:,2}; \boldsymbol{0}\right] + \left[\boldsymbol{0}; \boldsymbol{0}; 3M^{(3)}_{:,1}\right]. \tag{75}$$

In this construction, the information flow graphs from input 1 to the decision variable ($dv$) in each context for the two RNNs are shown in *Figure 7—figure supplement 3A*.

Although the connectivity matrices of the two networks are different (*Figure 7—figure supplement 3B*), when provided with the same input, the neuron activity of the $i$-th neuron in RNN1 is exactly the same as that of the $i$-th neuron in RNN2 at the same time point (*Figure 7—figure supplement 3C*). The statistical results of similarity between all neuron pairs are given in *Figure 7—figure supplement 3D*.

## Manually adding redundant structure and PEV of irrelevant activity (Figure 7—figure supplement 4)

### Redundant RNN (*Figure 7—figure supplement 4B and C*)

To see if the currently proposed method can work when there is a significant amount of neural activity variance irrelevant to the task, we manually added irrelevant neural activity into the trained RNNs (termed as redundant RNNs). Specifically, we randomly choice $K = 75$ trained vanilla RNNs (*Figure 7D–G*). To add irrelevant additional dimensions to the network without affecting the original function, for each RNN with N neurons, we build a larger RNN with $2N$ neurons equally divided into two populations. The input embeddings, readout vector, and the connectivity strengths between neurons within the first population (the first to the $N$-th neurons) are the same as the original RNN. As for the second population (the $N + 1$-th to the $2N$-th neurons), the input embeddings are sampled with Gaussian noise, with the standard deviation equal to that of the original RNN input embeddings and the readout vector is set to 0. The connectivity strengths within the second population are sampled with Gaussian noise, with the standard deviation chosen such that the average sum of square activity of the two populations is nearly equal. Moreover, there is no connectivity between the two populations. The resulting RNN, compared to the original network, introduces adding task-irrelevant neural activity (neural activity in the second population). At the same time, it performs the CDM task in the same way as the original network, with the identical selection vector modulation value.

### PEV of irrelevant activity (*Figure 7—figure supplement 4C*)

We hypothesize that the proportion of explained variance (PEV) in extra dynamical modes performs reliably for trained RNNs (*Figure 7G*) because task-irrelevant neural activity is minimal in these networks. To test this possibility, we conducted in-silico lesion experiments for the trained RNNs. The main idea is that if an RNN contains a large portion of task-irrelevant variance, there will exist a subspace (termed as task-irrelevant subspace) that captures this part of variance and removing this task-irrelevant subspace will not affect the network's behavior.

Based on this idea, we developed an optimization method to identify such a task-irrelevant subspace for any given RNN. The main idea is to find a lesion subspace such that removing all neural activity within this subspace does not affect the network's behavior, while capturing as much variance in neural activity caused by pulse inputs as possible. Specifically, for any given RNN, we constructed a new RNN with its connectivity matrix defined as $J^{'} = J\left(E - QQ^T\right)$, where $J$ is the connectivity matrix of the original RNN, $E$ is the identity matrix, and $Q$ is the orthogonal basis of the lesion subspace. Here, $Q$ is of shape $N \times r$, where $r$ is the dimension of the subspace ($r = 10$ in *Figure 7—figure supplement 4*). For the new RNN, the matrix $E - QQ^T$ removes all components of neural activity within the lesion subspace. We try to find the best $Q$ by minimizing:

$$\ell = \lambda\frac{(Z_{40} - \hat{Z}_{40})^2}{(\hat{Z}_{40})^2} + (1 - \frac{\|\hat{f}Q\|_2^2}{\|\hat{f}\|_2^2}) \tag{76}$$

where $z_{40}$ is the network's output at the last time step for pulse input at the beginning, $z_{40}$ is the output without lesion, $f$ is a $40 \times N$ matrix, indicating the neuron activities caused by a pulse input for the RNN without lesion, $\lambda = 10$ is a hyperparameter balancing the two parts. We only consider input

1 under condition one for simplicity. The first part ensures that after removing neural activity in the $Q$ subspace, the network's output in response to unit pulse remains unchanged. The second part aims to maximize the variance of neuron activity captured by lesion subspace. The orthogonal basis $Q$ is parameterized using the Cayley transform: $Q = \left(E + A - A^T\right)\left(E - A + A^T\right)^{-1}$. Matrix $A$ is optimized using gradient descent with the Adam optimizer with a learning rate of $10^{-5}$. The training process lasts for 2000 steps and the maximum of $\left\|fQ_2^2 \frac{}{\|f\|^2}\right.$ under the condition of $\frac{(Z_{40} - \hat{Z}_{40})^2}{(\hat{Z}_{40})^2} < 10^{-3}$ is defined as the network's PEV of irrelevant activity (**Figure 7—figure supplement 4C**).

# Acknowledgements

We thank Gouki Okazawa and Xuexin Wei for valuable discussions and comments on earlier versions of the manuscript. This work was supported by the Ministry of Science and Technology of China (STI2030-Major Project, 2021ZD0204105) and the National Natural Science Foundation of China (32271149).

# Additional information

### Funding

| Funder | Grant reference number | Author |
|---|---|---|
| Ministry of Science and Technology (China) | STI2030-Major Project, 2021ZD0204105 | Bin Min |
| National Natural Science Foundation of China | 32271149 | Bin Min |

The funders had no role in study design, data collection and interpretation, or the decision to submit the work for publication.

### Author contributions

Yiteng Zhang, Conceptualization, Software, Validation, Investigation, Visualization, Methodology, Writing – original draft, Writing – review and editing; Jianfeng Feng, Resources, Supervision, Writing – review and editing; Bin Min, Conceptualization, Resources, Supervision, Investigation, Visualization, Writing – original draft, Writing – review and editing

### Author ORCIDs

Yiteng Zhang ⓘ https://orcid.org/0009-0001-5192-521X
Jianfeng Feng ⓘ https://orcid.org/0000-0001-5987-2258
Bin Min ⓘ https://orcid.org/0000-0003-1006-9629

Reviewer #1 (Public review): https://doi.org/10.7554/eLife.103636.3.sa1
Reviewer #2 (Public review): https://doi.org/10.7554/eLife.103636.3.sa2
Author response https://doi.org/10.7554/eLife.103636.3.sa3

# Additional files

### Supplementary files

MDAR checklist

### Data availability

All code and models supporting the findings of this study have been deposited in GitHub (copy archived at **Zhang, 2025**). Additional materials are available from the corresponding author upon reasonable request.

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
