## [Editor Report · eLife Assessment]

This study provides an **important** set of analyses and theoretical derivations to understand the mechanisms used by recurrent neural networks (RNNs) to perform context-dependent accumulation of evidence. The results regarding the dimensionality and neural dynamical signatures of RNNs are **convincing** and provide new avenues to study the mechanisms underlying context-dependent computations. This manuscript will be of interest to a broad audience in systems and computational neuroscience.

---

## [Referee Report · Reviewer #1 (Public review)]

Summary:

This paper investigates how recurrent neural networks (RNNs) can perform context-dependent decision-making (CDM). The authors use low-rank RNN modeling and focus on a CDM task where subjects are presented with sequences of auditory pulses that vary in location and frequency, and they must determine either the prevalent location or frequency based on an external context signal. In particular, the authors focus on the problem of differentiating between two distinct selection mechanisms: input modulation, which involves altering the stimulus input representation, and selection vector modulation, which involves altering the "selection vector" of the dynamical system.

First, the authors show that rank-one networks can only implement input modulation, and that higher-rank networks are required for selection vector modulation. Then, the authors use pathway-based information flow analysis to understand how information is routed to the accumulator based on context. This analysis allows the authors to introduce a novel definition of selection vector modulation that explicitly links it to changes in the effective coupling along specific pathways within the network.

The study further generates testable predictions for differentiating selection vector modulation from input modulation based on neural dynamics. In particular, the authors find that: (1) A larger proportion of selection vector modulation is expected in networks with high-dimensional connectivity. (2) Single-neuron response kernels exhibiting specific profiles (peaking between stimulus onset and choice onset) are indicative of neural dynamics in extra dimensions, supporting the presence of selection vector modulation. (3) The percentage of explained variance (PEV) of extra dynamical modes extracted from response kernels at the population level can serve as an index to quantify the amount of selection vector modulation.

Strengths:

The paper is clear and well written, and it draws bridges between two recent important approaches in the study of CDM: circuit-level descriptions of low-rank RNNs, and differentiation across alternative mechanisms in terms of neural dynamics. The most interesting aspect of the study involves establishing a link between selection vector modulation, network dimensionality and dimensionality of neural dynamics. The high correlation between the networks' mechanisms and their dimensionality (Fig. 7d) is surprising since differentiating between selection mechanisms is generally a difficult task, and the strength of this result is further corroborated by its consistency across multiple RNN hyperparameters (Figure 7-figure supplement 1 and Figure 7-figure supplement 2). Interestingly, the correlation between the selection mechanism and the dimensionality of neural dynamics is also high (Fig. 7g), potentially providing a promising future avenue for the study of neural recordings in this task.

Weaknesses:

As acknowledged by the authors, the results linking selection vector modulation and dimensionality might not generalize to neural representations where a significant fraction of the variance encodes information unrelated to the task. Therefore, these tools might not be applicable to neural recordings or to artificial neural networks with additional high-dimensional activity unrelated to the task (e.g. RNNs trained to perform many other tasks).

---

## [Referee Report · Reviewer #2 (Public review)]

This manuscript examines network mechanisms that allow networks of neurons to perform context-dependent decision-making.

In a recent study, Pagan and colleagues identified two distinct mechanisms by which recurrent neural networks can perform such computations. They termed these two mechanisms input-modulation and selection-vector modulation. Pagan and colleagues demonstrated that recurrent neural networks can be trained to implement combinations of these two mechanisms, and related this range of computational strategies with inter-individual variability in rats performing the same task. What type of structure in the recurrent connectivity favors one or the other mechanism however remained an open question.

The present manuscript addresses this specific question by using a class of mechanistically interpretable recurrent neural networks, low-rank RNNs.

The manuscript starts by demonstrating that unit-rank RNNs can only implement the input-modulation mechanism, but not the selection-vector modulation. The authors then build rank three networks which implement selection-vector modulation, and show how the two mechanisms can be combined. Finally, they relate the amount of selection-vector modulation with the effective rank, ie the dimensionality of activity, of a trained full-rank RNN.

Strength:

- The manuscript is written in an obvious manner

- The analytic approach adopted in the manuscript is impressive

- Very clear identification of the mechanisms leading to the two types of context-dependent modulation

- Altogether, this manuscript reports remarkable insights on a very timely question

---

## [Author Response]

The following is the authors’ response to the original reviews

**Reviewer 1 (Public review):**
The first part of the manuscript is not particularly novel, and it would be beneficial to clearly state which aspects of the analyses and derivations are different from previous literature. For example, the derivation that rank-1 RNNs cannot implement selection vector modulation is already present in the Extended Discussion of Pagan et al., 2022 (Equations 42-43). Similarly, it would be helpful to more clearly explain how the proposed pathway-based information flow analysis differs from the circuit diagram of latent dynamics in Dubreuil et al., 2022.

We thank the reviewer for the insightful comments and providing us a good opportunity to better clarify the novelty of our work regarding the analyses and derivations. In general, as the reviewer pointed out, the major novelty of our work lies in explicitly linking selection mechanisms (proposed in Mante et al. 2013) with circuit-level descriptions of low-rank RNNs (developed in Dubreuil et al. 2022). This is made possible through a set of analyses and derivation integrating both linearized dynamical systems analysis (Mante et al., 2013) and the circuit diagram of latent dynamics (Dubreuil et al. 2022). Specifically, starting from rank-3 RNN models, we first derived the circuit diagram of latent dynamics (Eqs. 18 and 19) by applying the theory developed in Dubreuil et al. 2022. However, without further analysis, there is no explicit link between this latent dynamics and selection mechanism. In this manuscript, based on the line attractor assumption, we linearized the latent dynamics around the line attractor (Mante et al., 2013), which enabled us to explicitly solve the equation (from eq. 20 to eq. 27) and derive an explicit formula for the effective coupling of information flow (Fig. 5A). This formula of effective coupling strength supported an explicit pathway-based definition of selection vector modulation (Fig. 5) and selection vector (Fig. 6), the core result of this manuscript. Importantly, the same analysis can be extended to higher-order lowrank RNNs (Eqs. 47-55), suggesting the general applicability of our result. We have revised the manuscript to clearly state the novelty of our work. Please see Lines 292-294.

As such a set of analyses and derivation integrates many results from previous literatures, it naturally shared many similarities with previous results as the reviewer pointed out. Below, we compared our work with previous ones mentioned by the reviewer:

(1) For example, the derivation that rank-1 RNNs cannot implement selection vector modulation is already present in the Extended Discussion of Pagan et al., 2022 (Equations 42-43).

For this point, we totally agree with the reviewer that the derivation of rank-1 RNNs’ limitations in implementing selection vector modulation is not particularly novel. The reason why we started from rank-1 RNNs is because these RNNs are the simplest examples revealing the intriguing link between the connectivity property and the modulation mechanism and thereby serving as the ideal introduction for the subsequent in-depth discussion for general audiences. In the original manuscript, we cited the Pagan et al. 2023 note but may not make it explicit enough. As the reviewer pointed out that the derivation has been added into the latest version of Pagan et al. paper (Pagan et al. 2024), we now cite the Pagan et al. 2024 paper and make it clear that the derivation has been derived in Pagan et al. 2024. Please see Lines 186-188 in the main text.

(2) Similarly, it would be helpful to more clearly explain how the proposed pathway-based information flow analysis differs from the circuit diagram of latent dynamics in Dubreuil et al., 2022.

As we explained earlier, the latent dynamics in Dubreuil et al. alone did not provide an explicit link between circuit diagram and selection mechanisms. Our analysis go beyond the theory developed in Dubreuil et al. 2022 paper by integrating the linearized dynamical systems analysis (Mante et al. 2013), eventually providing a previously-unknown explicit link between circuit diagram and selection mechanisms.

With regard to the results linking selection vector modulation and dimensionality, more work is required to understand the generality of these results, and how practical it would be to apply this type of analysis to neural recordings. For example, it is possible to build a network that uses input modulation and to greatly increase the dimensionality of the network simply by adding additional dimensions that do not directly contribute to the computation. Similarly, neural responses might have additional high-dimensional activity unrelated to the task. My understanding is that the currently proposed method would classify such networks incorrectly, and it is reasonable to imagine that the dimensionality of activity in high-order brain regions will be strongly dependent on activity that does not relate to this task.

We thank the reviewer for this insightful comment. As what the reviewer suggested, we did more work to better understand the generality and applicability of the index proposed in the manuscript.

Firstly, to see if the currently proposed method can work when there is significant amount of neural activity variance irrelevant to the task, we manually added irrelevant neural activity into the trained RNNs (termed as redundant RNNs, see Methods for details, Lines 1200-1215). As expected, we found that for these redundant RNNs, the correlation between the proposed index and the proportion of selection vector modulation indeed disappeared (Figure 7-figure supplement 4B). In fact, in the original version of our manuscript, we presented an extreme example of this idea in our discussion, where we designed two RNNs with theoretically identical neural activity patterns—one relying purely on input modulation and the other on selection vector modulation (Figure 7-figure supplement 3). Therefore, for this extreme example, any activity-based index alone would fail to differentiate between these two mechanisms, suggesting the challenge of distinguishing different selection mechanisms when taskirrelevant neural activity is added.

Secondly, we asked why the proposed index works well for the trained RNNs, which is kind of surprising in the first place as the reviewer pointed out. One possibility is that for trained RNNs, the task-irrelevant neural activity is minimal. To test this possibility, we conducted in-silico lesion experiments for the trained RNNs. The main idea is that if an RNN contains a large portion of taskirrelevant variance, there will exist a subspace (termed as task-irrelevant subspace) that captures this part of variance and removing this task-irrelevant subspace will not affect the network’s behavior. Based on this idea, we developed an optimization method to identify such a task-irrelevant subspace for any given RNN (see Methods for details, Lines 1216-1244). The results show that in the originally trained RNNs, the identified task-irrelevant subspace can only explain a small portion of neural activity variance (Figure 7-figure supplement 4, panel C). As a control, when applying the same optimization method to the redundant RNNs, we found that the identified task-irrelevant subspace can explain a significantly larger portion of neural activity variance (Figure 7-figure supplement 4, panel C). Taken together, we concluded that the reason why the index works for trained RNNs is because the major variance of the neural activity of the network learned through backpropagation is task-relevant.

Therefore, this set of analyses provided an understanding why the proposed index works for trained RNNs and failed for the redundant RNNs. We have added this part of analyses in the Discussion part. See Lines 601-610. As the reviewer pointed out that it is highly likely that there exists taskirrelevant neural activity variance in high brain regions, the proposed index may not work well in neural recordings. With this understanding, we tone down the conclusion related to experimentally testable prediction in the main text (e.g., in Abstract and Introduction). We thank the reviewer again for helping us improve the clarity of our work.

Finally, a number of aspects of the analysis are not clear. The most important element to clarify is how the authors quantify the "proportion of selection vector modulation" in vanilla RNNs (Figures 7d and 7g). I could not find information about this in the Methods, yet this is a critical element of the study results. In Mante et al., 2013 and in Pagan et al., 2022 this was done by analyzing the RNN linearized dynamics around fixed points: is this the approach used also in this study? Also, how are the authors producing the trial-averaged analyses shown in Figures 2f and 3f? The methods used to produce this type of plot differ in Mante et al., 2013 and Pagan et al., 2022, and it is necessary for the authors to explain how this was computed in this case.

We thank the reviewer for the valuable comments. Yes, for proportion of selection vector modulation (Figure 7D and 7G) we employed the method used in Mante et al., 2013. For the trial-averaged analyses shown in Figures 2f and 3f, we followed a procedure used in Mante et al., 2013. In the revised version, we have added the relate information. See Lines 852-853 and 872-889. We thank the reviewer again for improving the clarify of our work.

I am also confused by a number of analyses done to verify mathematical derivations, which seem to suggest that the results are close to identical, but not exactly identical. For example, in the histogram in Figure 6b, or the histogram in Figure 7-figure supplement 3d: what is the source of the small variability leading to some of the indices being less than 1?

In Figure 6B, the two selection vectors are considered theoretically equivalent under the meanfield assumption. However, because the RNNs we use have a finite number of neurons, finite-size effects inevitably cause slight deviations from perfect equivalence.

To verify this, we generated rank-3 RNNs of different sizes in the experiment for Figure 6b (see the Supplementary section “Building rank-3 RNNs with both input and selection vector modulations”). Specifically, for a fixed number of neurons 𝑁, we independently sampled 𝛼, 𝛽 and 𝛾 from a Uniform(0,1) distribution and built an RNN with 𝑁 neurons based on the procedure as in Figure 5. We then computed the selection vector for the RNN in a given context (for example, context 1) in two ways:

(1) via linearized dynamical system analysis following Mante et al. (2013), producing the selection vector sc^classical^(2) using the theoretical derivation \begin{document}$s v=n_{d v}+\eta n_{i v_{1}}$\end{document}

**Author response image 1. sa3fig1:** cos angles for selection vectors computed using two methods in RNN with different size. Black bars indicate median values.

We repeated this process 1000 times for each 𝑁 and measured the cosine angle between these two selection vectors. As shown in Author response image 1, as 𝑁 increases, the cosine angles approach 1 more consistently, indicating that the two selection vectors become nearly equivalent in larger RNNs. Conversely, smaller RNNs display more pronounced finite-size effects, which accounts for indices slightly below 1.

**Reviewer 2 (Public review):**
The introduction could have been written in a more accessible manner for any non-expert readers.

We sincerely thank the reviewer for the constructive feedback on the introduction and have revised it accordingly.

**Reviewer #2 (Recommendations for the authors):**
The level of mastery of the low-rank framework is altogether impressive. I need however to point to a technical detail. The derivations of the information flow assume that the vectors m and vectors I are orthogonal (e.g. in Equation 14). This is not necessarily the case in trained networks, and Figure 2F suggests this is not the case in the trained rank 1 network. In that situation, the overlap between m and I leads to an additional term in the Equation going directly from the input to the output vector (see, e.g., Equation 15 in Beiran et al. Neuron 2023). In general, these kind of overlaps can contribute an additional pathway in higher rank networks too.

We thank the reviewer for the valuable comments. The derivations presented in Equation 14 do not actually require that the vectors 𝒎 and 𝑰 are orthogonal. Rather, our definition of the task variable differs slightly from the one in Beiran et al. (2023). Consider a rank-1 RNN with a single input channel:\begin{document}$$\displaystyle  \tau \frac{d \boldsymbol{x}}{d t}=-x+\frac{1}{N} \boldsymbol{m} \boldsymbol{n}^{T} \phi(\boldsymbol{x})+\boldsymbol{I} u .$$\end{document}

**Author response image 2. sa3fig2:** Difference of the definition of task variable with previous work. (A) Our definition of task variable. (B) Definition of task variable in Beiran et al. 2023.

As long as 𝒎 and 𝑰 are linearly independent, the state 𝒙(𝑡) can be uniquely written as a linear combination of the two vectors (Author response image 2):\begin{document}$$\displaystyle  \boldsymbol{x}(t)=\kappa_{d v}(t) \boldsymbol{m}+\kappa_{\text {inp }}(t) \boldsymbol{I},$$\end{document}

where \begin{document}$k_{d v}$\end{document} and \begin{document}$k_{i n p}$\end{document} are the task variables associated with 𝒎 and 𝑰, respectively. Substituting this expression into the dynamical equations yields:\begin{document}$$\displaystyle   \tau \frac{d \kappa_{d v}}{d t} &=-\kappa_{d v}+\frac{1}{N} \boldsymbol{n}^{T} \phi(\boldsymbol{x}), \\  \tau \frac{d \kappa_{i n p}}{d t} & =-\kappa_{i n p}+u$$\end{document}

Hence, there is no additional term directly linking the input to the output vector in our formulation. By contrast, in Beiran et al. (2023), the input vector 𝑰 is decomposed into components parallel (𝐼//) and perpendicular (𝑰-) to 𝒎, and the task variables \begin{document}$\kappa_{d v}^{\prime}, \kappa_{i n p}^{\prime}$\end{document} are defined as (Figure 4-figure supplement 3B):\begin{document}$$\displaystyle  \boldsymbol{x}(t)=\kappa_{d v}^{\prime}(t) \boldsymbol{m}+\kappa_{\text {inp }}^{\prime}(t) \boldsymbol{I}^{\perp}$$\end{document}

This leads to dynamics of the form:\begin{document}$$\displaystyle   \tau \frac{d \kappa_{d v}^{\prime}}{d t} &=-\kappa_{d v}^{\prime}+\frac{1}{N} \boldsymbol{n}^{T} \phi(\boldsymbol{x})+\frac{\boldsymbol{m}^{T} \boldsymbol{I}}{\boldsymbol{m}^{T} \boldsymbol{m}} u \\  \tau \frac{d \kappa_{i n p}^{\prime}}{d t} &=-\kappa_{i n p}^{\prime}+u $$\end{document}

thus creating an additional direct term from the input to the output vector under their definition.

The designed rank 3 network relies on a multi-population structure. This is explained clearly in the methods, but it could be stressed more in the main text to dispel the notion that higherrank networks may not need a multi-population structure to perform this task (cf Dubreuil et al 2022).

Thank you for the valuable comments. In the revised version, we emphasize this point by adding the following sentence: “our rank-3 network relies on a multi-population structure, consistent with the notion that higher-rank networks still require a multi-population structure to perform flexible computations (Dubreuil et al. 2022)”. See Lines 238-240.

(3) An important result in Pagan et al and Mante et al is that the line attractor direction is invariant across contexts. I believe this is explicitly enforced in the models studied here, but this could be made more clear. It would be interesting to discuss the importance of this constraint.

We thank the reviewer for the valuable comments. In our hand-crafted RNN examples (Figures 3– 6), we enforce the choice axis to be identical across the two contexts (Figure R4B). Even in the rank-1 example (Figure 2), where we analyze a trained RNN, the choice axis still shows a substantial overlap between the two contexts (Figure R4A). However, in the trained vanilla RNNs shown in Figure 7, when the regularization term is relatively small, the overlap in the choice axis between contexts is smaller (Figure R4C)—i.e., the line attractor direction shifts between different contexts.

**Author response image 3. sa3fig3:** Cosine angle between the choice axes in two contexts for different RNNs. (A) Rank-1 RNNs in Figure 2. (B) Rank-3 RNNs in Figure 3-6. (C) Vanilla RNNs in Figure 7.

Our theoretical framework can also accommodate situations where the direction of the choice axis changes. For instance, consider the rank-3 RNN in Figure 6, where the choice axis \begin{document}$\widetilde{\boldsymbol{m}}_{d v}$\end{document} is defined as \begin{document}$\tilde{\boldsymbol{m}}_{d v}=G \boldsymbol{m}_{d v}$\end{document} with 𝐺 being a diagonal matrix whose elements represent the slopes of each neuron’s activation function. Since these slopes can change across contexts, \begin{document}$\widetilde{\boldsymbol{m}}_{d v}$\end{document} itself can vary across contexts. Likewise, the input representation direction may be written as \begin{document}$\tilde{I}_{1}=G I_{1}$\end{document}, allowing both the choice axis and the input axis to adapt to the context. The selection vector is given by:\begin{document}$$\displaystyle  s=\frac{1}{N}\left\|\widetilde{\boldsymbol{m}}_{d v}\right\|_{2}\left(n_{d v}+E_{i v_{1} \rightarrow d v} n_{i v_{1}}+E_{i v_{2} \rightarrow d v} n_{i v_{2}}\right)$$\end{document}

Here, we no longer assume that \begin{document}$\widetilde{\boldsymbol{m}}_{d v}$\end{document} is context-invariant; rather, we only assume its norm \begin{document}$\left\|\tilde{\boldsymbol{m}}_{d v}\right\|_{2}$\end{document} remains the same across contexts. Under this weaker assumption, we still have\begin{document}$$\displaystyle   N \overline{\boldsymbol{s}} &=\left\|\widetilde{\boldsymbol{m}}_{d v}\right\|_{2}\left(\boldsymbol{n}_{d v}+\bar{E}_{i v_{1} \rightarrow d v} \boldsymbol{n}_{i v_{1}}+\bar{E}_{i v_{2} \rightarrow d v} \boldsymbol{n}_{i v_{2}}\right) \\  N \Delta \boldsymbol{s} &=\left\|\widetilde{\boldsymbol{m}}_{d v}\right\|_{2}\left(\Delta E_{i v_{1} \rightarrow d v} n_{i v_{1}}+\Delta E_{i v_{2} \rightarrow d v} n_{i v_{2}}\right) $$\end{document}

Substituting these into the equations yields the following expressions for input modulation and selection vector modulation:\begin{document}$$\displaystyle   \bmod _{i n p} &=\left\|\widetilde{\boldsymbol{m}}_{d v}\right\|_{2}\left(\Delta E_{i n p_{1} \rightarrow d v}+\Delta E_{i n p_{1} \rightarrow i v_{1}} \bar{E}_{i v_{2} \rightarrow d v}\right) \\  \bmod _{s e l} &=\left\|\widetilde{\boldsymbol{m}}_{d v}\right\|_{2}\left(\bar{E}_{i n p_{1} \rightarrow i v_{1}} \Delta E_{i v_{1}}\right) $$\end{document}

Figure 6B: it was not clear to me what exactly is plotted here.

We thank the reviewer for pointing out the missing explanation. In Figure 6B, we show the distribution of the cosine angles between two ways of computing the selection vector for randomly generated rank-3 RNNs. Specifically, We generate 1000 RNNs according to the procedure in Figure 5C, with each RNN defined by parameters 𝛼 , 𝛽 and 𝛾 independently sampled from a Uniform(0,1) distribution. For each RNN, we computed the selection vector for the RNN in a given context (e.g., context 1 or 2) in two ways:

(1) via linearized dynamical system analysis following Mante et al. (2013), producing the selection vector sv<supclassical (classical in Figure 6B),

(2) using the theoretical derivation \begin{document}$s v=n_{d v}+\eta n_{i v_{1}}$\end{document} (“our’s” in Figure 6B).

We repeated this process 1000 times and measured the cosine angle between these two selection vectors and plot the resulting distribution for context 1 (gray) and context 2 (blue) in Figure 6B. The figure shows that the computed selection vectors via the two methods are almost equal, as evidenced by the cosine angles clustering very close to 1.

We have revised it accordingly. See Lines 1135-1143.

In Figure 7, how was the effective dimension of vanilla RNNs controlled or varied? The metric used (effective dimension) is relatively non-standard, it would be useful to give some intuition to the reader about it.

We thank the reviewer for these valuable comments.

Controlling the effective dimension

When train vanilla RNNs, we included a regularization term in the loss function of the form\begin{document}$$\displaystyle  L_{r e g}=w_{r e g} \sum_{i j} J_{i j}^{2},$$\end{document}

where 𝑤536 is a regularization coefficient. By adjusting 𝑤536, we can influence the distribution of singular values of connectivity of 𝐽. When *wreg* is larger, the learned 𝐽 tends to have fewer large singular values, hence with lower effectivity dimension; when 𝑤536 is small, more singular values remain large, increasing the matrix’s effective dimension.

Definition and intuition: effective dimension

Consider a connectivity matrix 𝐽 with singular values \begin{document}$\sigma_{1} \geq \sigma_{2} \geq \cdots \geq \sigma_{N}$\end{document}. The matrix’s rank is the number of nonzero singular values. However, rank alone can overlook differences in how quickly those singular values decay. To capture this, we define the effective dimension as:\begin{document}$$\displaystyle  \mathrm{edim}(J)=\sum_{i=1}^{N} \frac{\sigma_{i}^{2}}{\sigma_{1}^{2}}$$\end{document}

Each term lies between 0 and 1, so the effective dimension satisfies:\begin{document}$$\displaystyle  0 \leq \operatorname{edim}(J) \leq \operatorname{rank}(J)$$\end{document}

When all nonzero singular values are equal, edim(𝐽) equals the matrix rank. But if some singular values are much smaller than others, effective dimension will be closer to 1. For example:

- 𝐽_1_ has nonzero singular values (1, 0.1, 0.01). Its effective dimension is 1.0101, indicating that most of the variance is captured by the largest singular value.

- 𝐽sub>0 has nonzero singular values (1, 0.8, 0.7). Its effective dimension is 2.13, which reflects that multiple singular values contribute significantly.

Hence, while both >𝐽_1_ and 𝐽sub>0 are rank-3 matrices, their effective dimensions highlight the difference in how each matrix distributes its variance.

We have added the intuition underlying this concept in Methods (see Lines 1135-1143). We thank the reviewer for improving the clarity of our work.

Eqs 19&21: n^T_r should be n^T_dv?

Thank you for point out this mistake. We have fixed it in the revised version.